# TD-M(PC)²: Improving Temporal Difference MPC Through Policy Constraint

## Abstract

Model-based reinforcement learning (MBRL) algorithms that integrate *model predictive control* with learned value or policy priors have shown great potential to solve complex continuous control problems. However, existing practice relies on online planning to collect high-quality data, resulting in value learning that is entirely dependent on off-policy experiences. Contrary to the belief that value learned from model-free policy iteration within this framework is sufficiently accurate and expressive, we found that severe value overestimation bias occurs, especially in high-dimensional tasks. Through both theoretical analysis and empirical evaluations, we identify that this overestimation stems from a structural policy mismatch: the divergence between the exploration policy induced by the model-based planner and the exploitation policy evaluated by the value prior. To improve value learning, we emphasize conservatism that mitigates *out-of-distribution* queries. The proposed method, TD-M(PC)², addresses this by applying a soft-constrained policy update—a minimalist yet effective solution that can be seamlessly integrated into the existing plan-based MBRL pipeline without incurring additional computational overhead. Extensive experiments demonstrate that the proposed approach improves performance over baselines by large margins, particularly in 61-DoF humanoid control tasks.

## 1 Introduction

Model-Based Reinforcement Learning (MBRL) leverages an explicit model of the environment's dynamics to achieve both high asymptotic performance and strong sample efficiency across a variety of sequential decision-making problems (Littman & Moore, 1996; Janner et al., 2019; Chua et al., 2018; Schrittwieser et al., 2020). In continuous control domains, a particularly popular instantiation of MBRL is plan-based MBRL, where a trajectory optimizer, most often sampling-based Model Predictive Control (MPC), uses the learned model to perform short-horizon rollouts (Sikchi et al., 2022; Hansen et al., 2022). These methods incorporate a value prior acquired through RL methods as the terminal cost, enabling the planner to focus computational effort on near-term predictions while relying on the value function to estimate long-term returns beyond the planning horizon. This hybrid strategy not only reduces the planner's sensitivity to corresponding model error over long horizons but also accelerates exploration by rapidly covering the state space through parallel trajectory sampling (Lowrey et al., 2018).

Ideally, a value prior that accurately evaluates a near-optimal policy is required to provide an unbiased objective for the MPC planner. A natural choice is to learn this prior via model-free actor–critic methods (Sutton & Barto, 2018) as in TD-MPC (Hansen et al., 2022). While the online planner is leveraged for exploration and data collection, a nominal policy (exploitation policy) and the corresponding value prior are iteratively learned through policy iteration. However, this introduces a mismatch between exploration and exploitation policies. Thus, value learning must contend with heterogeneous experience, which causes an extreme form of the off-policy issue. This raises the critical question: Can plan-based MBRL algorithms effectively exploit such data for value learning, especially in high-dimensional environments?

Despite strong performance on low- and medium-dimensional benchmarks, state-of-the-art plan-based MBRL methods may struggle in high-dimensional tasks (Sferrazza et al., 2024). Through theoretical and empirical evidence, we identify persistent value overestimation that originates from

off-policy exploration as a core bottleneck. Under the scope of approximate policy iteration (API) (Sutton & Barto, 2018; Munos, 2003), the structural policy mismatch inherently compounds with approximation errors. Such divergence leads to a distribution shift, making the value function bootstrapped on out-of-distribution actions. Consequently, the approximation error accumulates over iterations, and value overestimation is left unfixed and enlarged. Although related to the classical off-policy issue in model-free RL (Thrun & Schwartz, 2014; Sutton & Barto, 2018; Fujimoto et al., 2018; Van Hasselt et al., 2018), this particular case is more problematic and resembles the distribution shift issue in offline RL literature (Levine et al., 2020; Fujimoto et al., 2019): the behavior policy remains misaligned with the nominal policy over practical training time.

To address these challenges, we introduce $\underline{T}$emporal $\underline{D}$ifference Learning for $\underline{M}$odel $\underline{P}$redictive $\underline{C}$ontrol with $\underline{P}$olicy $\underline{C}$onstraint, TD-M(PC)$^2$, a simple but effective extension of the TD-MPC framework that better exploits fully off-policy data collected from online planning. By incorporating a distribution-constrained conservative policy update, TD-M(PC)$^2$ learns a policy prior that remains close to the behavior policy, thereby mitigating out-of-distribution queries that exacerbate value approximation errors. Practically, TD-M(PC)$^2$ can be implemented atop TD-MPC2 Hansen et al. (2023) with fewer than ten lines of modification and introduces negligible additional computational overhead. We evaluated our method on the DeepMind Control Suite (Tassa et al., 2018) and HumanoidBench (Sferrazza et al., 2024), where it achieved over a 100% improvement compared to the baseline on high-dimensional humanoid control tasks. The contribution of this paper can be summarized as follows: 1) We uncover and quantify a previously overlooked value overestimation issue in plan-based MBRL and demonstrate theoretically how this issue could be a bottleneck for continuous control problems with a high-dimensional state-action space. 2) We propose a simple yet efficient algorithm within the plan-based MBRL framework that addresses the value overestimation; 3) We demonstrate the superiority of our method on diverse high-dimensional continuous control tasks.

## 2 PRELIMINARIES

Continuous control problems can be defined as a Markov decision process (MDP) (Bellman, 1957) represented by tuple $\mathcal{M} = (\mathcal{S}, \mathcal{A}, \rho, \rho_0, r, \gamma)$, with state space $\mathcal{S}$, action space $\mathcal{A}$, transition of states $\rho(s'|s, a)$, initial state distribution $\rho_0$, reward function $r(s, a)$ and discount factor $\gamma \in (0, 1]$. Under the assumption of infinite horizon, the objective is to learn a optimal policy $\pi^*$ that maximizes discounted cumulative reward $J(\pi) = \mathbb{E}_{\tau^\pi}\left[\sum_{t=0}^{T} \gamma^t r(s_t, a_t)\right]$, where $\tau^\pi$ is a trajectory sampled from policy $\pi : \mathcal{S} \to \Delta(\mathcal{A})$, $\Delta(\mathcal{A})$ denotes a distribution over the action space.

The actor–critic framework combines policy and value learning by maintaining an explicit policy (the actor) alongside a state–action value function (the critic) (Sutton & Barto, 2018). This setup allows an agent to improve a target policy using evaluations derived from off-policy data. In particular, they can by considered as approximate policy iteration (API) (Munos, 2003; Agarwal et al., 2019) that aiming at acquiring an approximation $\hat{Q}(s, a)$ of the optimal state–action value function $Q^*(s, a) = \mathbb{E}_{\tau^{\pi^*}}[\Sigma_{t=0}^{\infty} \gamma^t r_t | s_0 = s, a_0 = a]$. It proceeds by iteratively solving policy evaluation $Q^k = \arg\min_Q \|Q - \mathcal{T}^{\pi_k} Q\|$ and policy improvement $\pi_{k+1} = \arg\max_\pi \mathbb{E}_s \mathbb{E}_{a \sim \pi}[Q^k(s, a)]$, where the Bellman operator is given by:

$$\mathcal{T}^{\pi_k} Q(s, a) = r + \alpha \mathbb{E}_{s' \sim p(\cdot|s,a), a' \sim \pi_k(\cdot|s')}[Q(s', a')] \tag{1}$$

Model-based RL explicitly leverages knowledge about the dynamics by learning a latent world model and searching for the optimal policy within that model. Then, the resulting planner policy is leveraged for exploration to collect high-quality data. However, planner performance at convergence could be compromised by compounding model error (Bhardwaj et al., 2020b). Thus, a widely adopted practice is to perform short-horizon planning with a value function $\hat{V}(s) = \mathbb{E}_{a \sim \pi(\cdot|s)}[Q(s, a)]$ as the terminal cost. A closed-loop control policy is acquired through trajectory optimization methods such as Model Predictive Path Integral (MPPI) (Williams et al., 2016; 2017). At each time step, parameters $\mu^*$ and $\sigma^*$ of a multivariate Gaussian are iteratively optimized

to maximize the expected return:

$$\mu^*, \sigma^* = \arg\max_{\mu,\sigma} \mathbb{E}_{(a_t,a_{t+1},\ldots,a_{t+H})\sim\mathcal{N}(\mu,\sigma^2)}[G(s_t)]$$

$$G(s_t) = \sum_{h=t}^{H-1} \gamma^h r(s_h, a_h) + \gamma^H \hat{V}(s_{t+H}) \tag{2}$$

$$\text{s.t.} \quad s_{t+1} = d(s_t, a_t)$$

where the update rule of the parameters $\mu, \sigma \in \mathbb{R}^{H \times m}$ can be interpreted as solving a stochastic optimal control problem via mirror gradient descent (Okada & Taniguchi, 2018; Chua et al., 2018). We conclude the full pipeline of the inference-time planning in the Appendix. Under a receding-horizon scheme, only the first action $a_t \sim \mathcal{N}(\mu_t^*; \sigma_t^{*2}\mathbf{I})$ of the planned trajectory is executed, and the subsequent planning horizon is warm-started by shifting the optimized mean $\mu_{t+1:H}^*$ forward.

**Notation.** We consider model-based RL (MBRL) with a planner that executes an $H$-step looka-head policy in the environment (Chua et al., 2018; Sikchi et al., 2022; Hansen et al., 2022). We denote this *planner policy* by $\pi_H$ (with $H$ as planning horizon), and a *nominal* policy by $\pi$ with corresponding value functions $V^\pi$ and $Q^\pi$ (Sutton & Barto, 2018). The policy iteration is indexed with a subscript,e.g., $\pi_k$ and $\pi_{H,k}$ stands for the nominal policy and the planner policy at iteration $k$. The interaction under $\pi_{H,k}$ appends transitions to a replay buffer $\mathcal{D}$. We denote $\mu$ for the *behavior policy* that induces the data distribution in $\mathcal{D}$; in practice, $\mu$ is the mixture over historical planner policies recorded in the buffer (i.e., the off-policy data-generating process).

## 3 Addressing Value Overestimation in Plan-Based MBRL

In this section, we demonstrate that value overestimation, driven by a policy mismatch between the MPC planner's H-step lookahead policy $\pi_H$ and the nominal policy used in value learning $\pi$, could be a bottleneck for plan-based MBRL. First, we analyze how value-approximation error affects planning quality under approximate policy iteration. Next, we empirically validate that value overestimation persists and even amplifies over iterations due to out-of-distribution bootstrapping. Finally, we provide a theoretical explanation of value overestimation and its relation to policy mismatch, motivating our conservative exploitation remedy.

### 3.1 Value Approximation and Planner Performance

Balancing model error and value error is crucial in MBRL. The effectiveness of MPC planners in approximating near-optimal solutions to the infinite-horizon MDP largely relies on a value that accurately reflects the optimal policy's long-term performance, especially considering that we favor short-horizon planning to ensure computation efficiency. Given that the value prior is typically learned via actor-critic methods, we quantify this dependency within the framework of approximate policy iteration (API). Specifically, Theorem 3.1 quantifies the suboptimality introduced by finite-horizon planning in the presence of approximation errors. A detailed proof is provided in Appendix B.7.

**Theorem 3.1** (Planner Performance Bound). *Assume at the $k$-th iteration, the nominal policy $\pi_k$ is acquired through API and the resulting planner policy is $\pi_{H,k}$. Denote $\epsilon_k$ as the approxima-tion error $\|\hat{V}_k - V^{\pi_k}\|_\infty$ of the learned value function $\hat{V}$. Also denote approximation error (w.r.t. TV-divergence) for dynamics model $\hat{\rho}$ as $\epsilon_m = \max_{s,a} D_{TV}(\rho(\cdot|s_t, a_t)\|\hat{\rho}(\cdot|s_t, a_t))$, planner sub-optimality as $\epsilon_p$. Finally, assume the reward function $r$ is bounded by $[0, R_{max}]$ and $\hat{V}$ is upper bounded by $V_{max}$, then the performance of $\pi_{H,k}$ is bounded w.r.t. the optimal policy as:*

$$\limsup_{k\to\infty} |V^* - V^{\pi_{H,k}}| \leq \limsup_{k\to\infty} \frac{2}{1-\gamma^H}\left[C(\epsilon_{m,k}, H, \gamma) + \frac{\epsilon_{p,k}}{2} + \frac{\gamma^H(1+\gamma^2)}{(1-\gamma)^2}\epsilon_k\right] \tag{3}$$

*while $C$ is defined as:*

$$C(\epsilon_m, H, \gamma) = R_{max}\sum_{t=0}^{H-1}\gamma^t t\epsilon_m + \gamma^H H\epsilon_m V_{max} \tag{4}$$

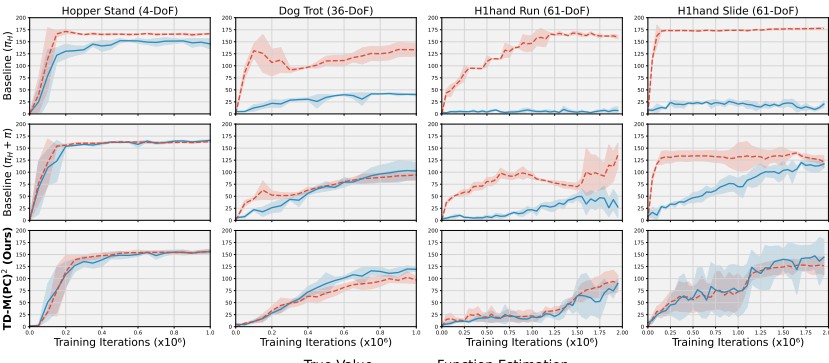

Figure 1: **Measuring overestimation bias in the value estimates of TD-MPC2**. We estimate the true value through Monte Carlo sampling. Specifically, we calculate the average discounted return over 100 episodes following the nominal policy $\pi$. The results are averaged over three seeds with UTD=1; The first row shows value estimates of vanilla TD-MPC2 with all data collected by the planner policy $\pi_H$; The second row shows TD-MPC2 with a probability of 0.5 to use the nominal policy $\pi$ for exploration; The third row shows value error of the proposed method with conservative exploitation. This suggests that the value overestimation is rooted in a mismatch between the exploration and exploitation policies.

Notably, Theorem 3.1 indicates that the planning procedure allows $\pi_H$ to mitigate its reliance on value accuracy by at least a factor of $\gamma^{H-1}$ compared to a greedy policy.[1] If we assume model error and planner suboptimality are insignificant or less important (e.g., short planning horizon), then converging value approximation error, $\epsilon_k$, guarantees converging planning performance. This begs an obvious question: As task complexity increases, particularly in environments with a high-dimensional state-action space, does the current framework yield a small value approximation error in practice?

## 3.2 EMPIRICAL EVIDENCE FOR VALUE OVERESTIMATION

While value approximation error is negligible in low-dimensional tasks, it could still be significant for complex tasks with a high-dimensional state-action space. The first row of Figure 1 illustrates value approximation error $\mathbb{E}_{s_0 \sim \rho_0}[\hat{V}(s_0) - V^\pi(s_0)]$ of TD-MPC2 in four distinct continuous control tasks from DMControl (Tassa et al., 2018) and HumanoidBench (Sferrazza et al., 2024), where the function estimation is given by $\hat{V} = \mathbb{E}_\pi[\hat{Q}]$: `Hopper-Stand` ($\mathcal{A} \in \mathbb{R}^4$, $\mathcal{O} \in \mathbb{R}^{15}$, 15% error), `Dog-Trot` ($\mathcal{A} \in \mathbb{R}^{38}$, $\mathcal{O} \in \mathbb{R}^{223}$, 231% error), `h1hand-run-v0` ($\mathcal{A} \in \mathbb{R}^{61}$, $\mathcal{O} \in \mathbb{R}^{151}$, 2159% error), `h1hand-slide-v0` ($\mathcal{A} \in \mathbb{R}^{61}$, $\mathcal{O} \in \mathbb{R}^{151}$, 746% error). More examples can be found in Figure 7. Notably, even though the planner performs poorly and collects trajectories with low return, value overestimation can still be large. This suggests that the approximation error is not due to overfitting high-value data generated by the planner, but rather an off-policy issue (Thrun & Schwartz, 2014; Sutton & Barto, 2018; Fujimoto et al., 2018) that bootstrapping, off-policy, and function approximation forms a troublesome "deadly triad" (Van Hasselt et al., 2018). In this case, Theorem 3.1 will no longer guarantee good performance for $\pi_H$.

Intuitively, approximation error arises when the behavior policy diverges from the nominal policy being evaluated by the value function. The planner, with full access to the learned model, yields a policy that behaves distinctly and potentially is superior to the nominal policy. This fully off-policy setting incurs an apparent distribution shift. Consequently, the function approximation error remains unfixed and increases as the value function is repeatedly queried with *out-of-distribution* data during bootstrapping.

To verify that this distribution shift drives value overestimation, we conduct a comparative experiment. We consider a variant of TD-MPC2 that periodically injects rollouts collected by the current nominal policy, $\pi$, into the replay buffer. Then we evaluate the value overestimation bias of this variant against the baseline. By mixing a fraction of on-policy data with the planner's off-policy tra-

---

[1]See detailed explanation in Appendix B.6

jectories, we relax the distribution shift and expect a corresponding reduction in approximation error. The empirical observation in Figure 1 meets our expectation, as value overestimation is significantly reduced on high-dimensional humanoid tasks.

## 3.3 ERROR ACCUMULATION

While the policy mismatch in TD-MPC delays the correction of value overestimation bias, one might expect that, given sufficient training, the agent would eventually visit overestimated regions and rectify the errors. Indeed, when the scale of value overestimation is large enough, the planner policy $\pi_H$ can also be approximately considered as a greedy policy of the value prior. Thus, it will explore the regions assigned erroneous large values more frequently until the value function corrects itself. For a low-dimensional task, this aligns with empirical observation. However, we argue that this self-correction could be difficult because value approximation errors not only propagate across states (Fujimoto et al., 2018) but also accumulate through policy iteration, making it highly troublesome for complex, high-dimensional tasks. We first quantify approximation error accumulation in the following theorem.

**Theorem 3.2** (TD-MPC Error Accumulation). *Assume $\pi_{H,k}$ outperforms $\pi_k$ with performance gap $\delta_k = \|V^{\pi_{H,k}} - V^{\pi_k}\|_\infty$. Denote value approximation error $\epsilon_k$, model error $\epsilon_{m,k}$, planner sub-optimality $\epsilon_p$ as defined before. Also let the reward function $r$ be bounded by $[0, R_{max}]$, then the following uniform bound of the performance gap holds:*

$$\delta_k \leq \frac{1}{1-\gamma^H}\left[2C(\epsilon_{m,k-1}, H, \gamma) + \epsilon_{p,k-1} + (1+\gamma^H)\delta_{k-1} + \frac{2\gamma(1+\gamma^{H-1})}{1-\gamma}\epsilon_{k-1}\right] \quad (5)$$

*where $C$ is defined in equation equation 4. We defer the complete proof to Appendix B.8.*

Note that the upper bound is quite loose due to the usage of the infinity norm. Nonetheless, the direct takeaway of Theorem 3.2 is that we can always expect a relatively large performance gap between the $H$-step lookahead policy and the nominal policy due to the accumulating approximation error. Consequently, a large value gap also indicates a larger policy mismatch.

**Theorem 3.3** (Distribution shift). *Given policies $\pi, \pi' \in \Pi : S \rightarrow \Delta(A)$, suppose the reward is upper bounded by $R_{max}$, then we have the TV-divergence of two visitation distributions lower bounded by the performance gap as follows. Proof can be found in Appendix B.4.*

$$D_{TV}(p^\pi(s,a)\|p^{\pi'}(s,a)) \geq \frac{1-\gamma}{2R_{max}}|J^\pi - J^{\pi'}| \quad (6)$$

As a result of a value error, the region corresponding to the nominal policy $\pi$ is underrepresented in the buffer, exacerbating the distributional shift rather than resolving it. Consequently, the enlarging distribution shift also leads to an extrapolation error of the value function, leaving a non-converging approximation error. Further discussion on the Theorems can be found in Appendix E.

Extrapolation errors had been well articulated in offline RL studies (Kumar et al., 2019; Peng et al., 2019; Levine et al., 2020). For model-free algorithms in online RL where exploration and exploitation policies align closely, with stabilizing methods applied to value learning (Fujimoto et al., 2018; Anschel et al., 2017; Lan et al., 2020), the overestimation is expected to be prevented, and even underestimation could take place (Hasselt, 2010). However, the policy mismatch brought by the planner largely affects value learning and distinguishes it from model-free off-policy learning.

**To summarize**, although the $H$-step lookahead policy is theoretically less sensitive to value approximation errors, a substantial amount of them are introduced and accumulate over training time due to policy mismatch. As a result, naively applying standard actor-critic methods fails to fully exploit the potential of combining model-based optimization and temporal-difference learning. These findings motivate our solution for better value learning.

# 4 IMPROVING VALUE LEARNING IN MBRL THROUGH A MINIMALIST APPROACH

To improve value learning in plan-based MBRL, we must address the challenge of exploiting planner-generated off-policy data. One natural remedy is to actively mitigate distribution shift by

incorporating the nominal policy into data collection (e.g., the method presented in Section 3.2). However, it empirically demands a large fraction of nominal policy rollouts to eliminate overestimation, which hinders the performance by reducing the planner's contribution. An alternative is to impose constraints on the planner itself, ensuring the exploration policy remains close to the prior, as in previous works (Sikchi et al., 2022; Grill et al., 2020). Yet this approach may stifle exploration in trajectory space. In contrast, we aim to enhance the exploitation of off-policy data with minimal effort. Inspired by recent progress in offline RL, where fully off-policy data is effectively leveraged via conservative updates, we consider integrating conservatism to leverage planner-collected off-policy data efficiently. In the following sections, we present **TD-M(PC)$^2$**, a framework that marries model-based exploration with conservative exploitation.

**Core Components.** We jointly learn encoder $z = h(s, e)$, latent dynamics $z' = d(z, a, e)$, reward function $\hat{r} = R(z, a, e)$, nominal policy $\hat{a} = \pi(z, e)$, and action value function $\hat{q} = \hat{Q}(z, a, e) \approx Q^\pi(z, a, e)$. where **z** is the latent state representation and $e$ is a learnable task embedding. Accordingly, we efficiently perform inference-time planning in the latent state-space. Specifically, $h, d, R, Q$ are jointly trained through the following loss:

$$\mathcal{L} \doteq \mathbb{E}_{(s,a,r,s')_{0:H}} \Big[ \sum_{t=0}^{H} \gamma^t \Big( \|d(z_t, a_t, e) - sg(h(s'_t))\|_2^2 + CE(\hat{r}_t, r_t) + CE(\hat{q}_t, q_t) \Big) \Big], \quad (7)$$

where the target $q_t$ is generated by bootstrapping nominal policy $\pi$ (refer to Equation (9)), and $sg$ is the stop-gradient operator. $\pi$ is a stochastic Tanh-Gaussian policy trained with the maximum Q objective in a model-free manner. During inference, $\pi$ selects actions at the terminal state, resulting in value estimation as $\hat{V}(z_{t+H}) = \mathbb{E}_{a_{t+H} \sim \pi(z_{t+H}, e)}[\hat{Q}(z_{t+H}, a_{t+H}, e)]$.

**Policy Iteration with Reduced OOD Query.** To limit out-of-distribution queries during bootstrapping, while avoiding overly pessimistic value collapse Nakamoto et al. (2024), we adopt a *soft, state-conditional divergence constraint* to the behavior policy $\mu$ (the mixture that induces the replay buffer $\mathcal{D}$), together with standard off-policy evaluation. At iteration $k$, we update the policy by

$$\pi_{k+1} \in \arg\max_{\pi} \mathbb{E}_{s \sim d_{\mathcal{D}}} \Big[ \mathbb{E}_{a \sim \pi(\cdot|s)} Q_k(s, a) - \beta D\big(\pi(\cdot \mid s) \,\|\, \mu(\cdot \mid s)\big) \Big], \quad (8)$$

where $D$ is a divergence (e.g., forward/reverse Kullback–Leibler divergence) and $d_{\mathcal{D}}$ is the empirical state distribution in replay. This BRAC-style (Wu et al., 2019) objective keeps $\pi_{k+1}$ *close to the data support* while still improving expected action values, in contrast to trust-region methods that constrain $\mathrm{KL}(\pi_{k+1} \,\|\, \pi_k)$ between successive policies (Schulman, 2015; Abdolmaleki et al., 2018). For evaluation, we apply the off-policy Bellman operator under the target policy $\pi_k$ and fit $Q_{k+1}$ from transitions sampled off-policy from $\mathcal{D}$:

$$\big(\mathcal{T}^{\pi_k} Q\big)(s, a) = r(s, a) + \gamma \, \mathbb{E}_{s' \sim \rho(\cdot|s,a)} \, \mathbb{E}_{a' \sim \pi_k(\cdot|s')} \big[ Q(s', a') \big]. \quad (9)$$

While the framework permits different divergence measures, we adopt a pragmatic instantiation prioritizing performance and simplicity. At a higher level, we consider the reverse KL-divergence as the measurement. In practice, we do not maintain a density estimate for the behavior policy, but store the mean and standard deviation for $\pi_{H,k}$ (multi-variant Gaussian) alongside the standard transition. We leverage a surrogate objective $\mathbb{E}_{\bar{\pi}_H \sim \mathcal{D}}[\log \bar{\pi}_H]$ as the lower bound of $\log \mu$.

To be noticed, unlike offline RL, exploration is essential under our setting, and overly conservative or excessive constraints could harm performance. Thus, we further provide design choices to enable "optimistic-conservatism". First, similar to practices in MaxEnt-RL (Haarnoja et al., 2018; 2017) we add entropy regularization to policy learning. Besides this widely adopted approach, we further loosen the constraint by not enforcing the conservative term until the running percentile $S_q$ (Eq. 11) exceeds a certain threshold $s$.[2] The overall policy loss yields the following.

$$\mathcal{L}_\pi = - \mathbb{E}_{s,\mu \sim \mathcal{B}, a \sim \pi} \left[ Q(s, a)/S_q + \beta \log \mu(a|s)/S_q - \alpha \log \pi(a|s) \right], \quad (10)$$

$$S_q = \mathrm{EMA}\left( \max\{\mathrm{Per}(Q, 95) - \mathrm{Per}(Q, 5), 1\}, \xi \right) \quad (11)$$

where we follow (Hansen et al., 2022; 2023) to scale the training loss by $S_q$ of the value function to improve stability. We summarize the overall pipeline as Algorithm 1.

---

[2] More intuition and empirical analysis on this hyperparameter can be found in Appendix D

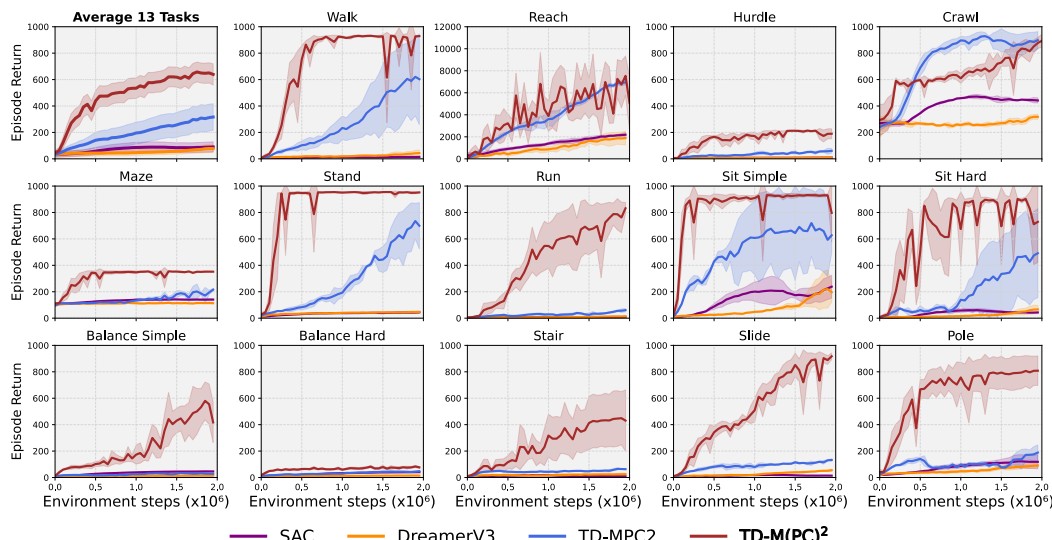

Figure 2: **Performance on Humanoid-Bench Locomotion Suite.** We report mean evaluation performance (Average episode return) and 95% CIs across 14 humanoid locomotion tasks. We omit `Reach-v0` in the average result due to the distinct reward scale.

Through experimental evaluation, we seek to answer the following *key questions*: How well does our approach perform on continuous control tasks, especially in environments of high-dimensional state-action space? Does our method reduce value approximation error and achieve accurate policy evaluation? Which component is critical to performance?

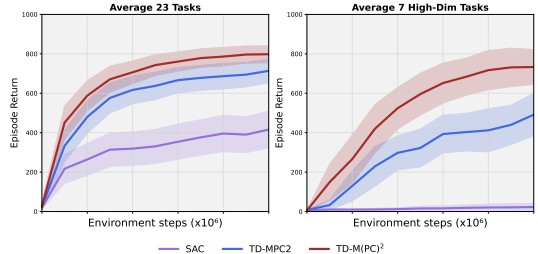

Figure 3: **Performance on DMControl suite.** We report mean evaluation performance and 95% CIs over 3 seeds. Left: Average performance on all 16 tasks; Right: Average performance on 7 high-dimensional tasks

**Benchmarks.** We evaluate our method across diverse tasks from HumanoidBench Sferrazza et al. (2024) and the DeepMind Control Suite (DMControl) Tassa et al. (2018). *Humanoid-Bench* is a standardized suite for humanoid control. We test all 14 locomotion tasks in `h1hand-v0` locomotion suite, which requires whole-body control of a humanoid robot with dexterous hands and a 61-dimensional action space. Additionally, we benchmark our method on 23 tasks from **DMControl**, including 7 high-dimensional tasks: `dog` (36-DoF) and `humanoid` (21-DoF).

**Baselines.** We mainly compare our method against TD-MPC2 (Hansen et al., 2023) and Soft Actor-Critic (Haarnoja et al., 2018). For Humanoidbench, we also include popular model-free and model-based algorithms, PPO (Schulman et al., 2017), DreamerV3 (Hafner et al., 2023). We leverage the implementation of SAC and PPO from StableBaseline3 (Raffin et al., 2021). We report baseline results on HumanoidBench from author-provided scores.[3] We implement TD-M(PC)$^2$ on top of the official repository of TD-MPC2[4] in PyTorch, ensuring all model architectures are identical for a fair comparison. Due to the space limit, please refer to Section D for further details.

## 4.1 IMPROVED VALUE LEARNING

We first assess the value overestimation bias of the proposed method using the same evaluation procedure as described in Section 3.2. As illustrated in the third row of Figure 1, our method

---

[3] https://github.com/carlosferrazza/humanoid-bench
[4] https://github.com/nicklashansen/tdmpc2

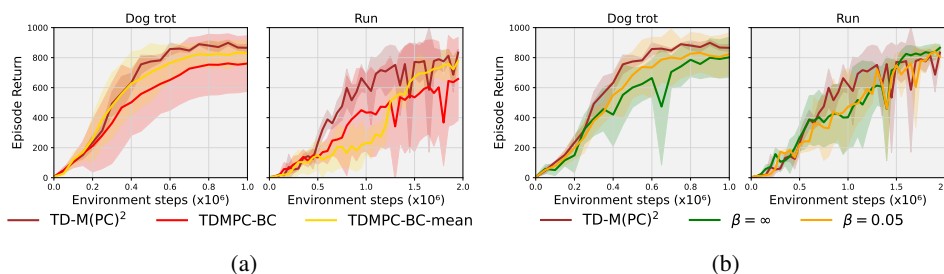

Figure 4: **Ablation study.** (a) Different regularization formulation yield close performance(b) Proposed method is not sensitive to $\beta$.

enables the learned value function to align better with the truth of the ground, reducing persistent bias in the estimation of values. This empirical result justifies our theoretical analysis. Moreover, the improvement in value accuracy is especially significant for high-dimensional humanoid tasks, where the baseline yields a persistent and non-converging bias. We will show that this improvement in value learning ultimately results in improved performance of the resulting H-step look-ahead policy.

### 4.2 BENCHMARK PERFORMANCE

We conducted our experiments across three random seeds for DMControl and five random seeds for HumanoidBench, with a total of 2M environmental steps and an up-to-date ratio (UTD) of one. We adopt the same settings for shared hyperparameters reported in the TD-MPC2 paper for our algorithm without task-specific tuning. A comprehensive list of hyperparameter settings for reproducibility and transparency in Table 1. For HumanoidBench, as demonstrated in Figure 2, our method consistently outperforms the baseline by a large margin for most tasks. In general, TD-M(PC)$^2$ tends to act with caution, while the baseline demonstrates an exaggerated motion corresponding to its overestimated value. Consequently, significant improvements are achieved in locomotion tasks, where the humanoid benefits more from performing stable and consistent motions.

We also evaluate our approach on some of the most challenging tasks in the DMControl suite, as in Figure 3, with comprehensive training curves presented in Figure 6 and Figure 5. On average, our method slightly outperforms the baseline. Notably, significant improvements are observed in three `dog` tasks, whereas the performance on `Humanoid` tasks, which feature a slightly smaller action space, remains comparable to the baseline. Complete training curve on DMControl and task visualization can be found in Section E.

### 4.3 ABLATION STUDY

To decide which formulation is the most suitable to address conservatism in policy learning, we compared the log-likelihood objective `TD-M(PC)`$^2$ to the behavior clone objective `TDMPC-BC` and behavior clone with respect to the mean of the behavior policy`TDMPC-BC-mean`. The log-likelihood objective shows the best performance in Figure 4a. While the BC objective is theoretically equivalent to the log-likelihood objective, it is less stable in practice. We also studied different ways to introduce conservatism, such as advantage-weighted regression (Peng et al., 2019).

To access sensitivity to $\beta$, we evaluate two variants of TD-M(PC)$^2$: a mildly regularized version with $\beta = 0.05$ and an extremely conservative variant that directly applies behavior cloning (BC) for policy updates $beta = \infty$. These variants are tested on two high-dimensional continuous control tasks. As shown in Figure 4b, all three variants achieve similar performance, suggesting that the framework is not sensitive to $\beta$. Notably, the BC variant performs nearly on par with the others despite the actor only imitating stale planner behavior. This indicates that reducing out-of-distribution actions is a key factor. Due to limited space, we postpone more ablation results on other design choices in Section E.

## 5 RELATED WORK

**Model-based RL.** Recent advances in MBRL aim to balance scalability and adaptability by integrating strengths from both model-free and model-based paradigms. Dyna-Q Sutton (1991) considers using simulated rollouts from a learned model to augment real-world experience, often referred to as "background planning" Hamrick et al. (2020). Janner et al. (2019) provides a theoretical guarantee of monotonic policy improvement with model-generated data. Katsigiannis & Ramzan (2017); Hafner et al. (2020; 2023) further introduce latent world models for high-dimensional tasks, e.g., visual control. Planning algorithms such as Model Predictive Control (MPC) and Monte Carlo Tree Search (MCTS) can explicitly exploit model knowledge to acquire a superior control policy. Planner is critical to policy learning, it can be applied to high-quality data acquisition Lowrey et al. (2018); Hansen et al. (2022), inference-time planning Chua et al. (2018); Hafner et al. (2019), or provide expert learning signal Schrittwieser et al. (2020); Bhardwaj et al. (2020b); Ye et al. (2021); Wang et al. (2024). Schrittwieser et al. (2020) achieves trust-region policy improvement through combining MCTS with policy and value prior Grill et al. (2020). Contrary to this, we do not rely on expert iteration Ye et al. (2021); Wang et al. (2025) for policy improvement, but leverage past behavior policy to enforce conservatism and improve exploitation.

**Off-policy Learning and Value Approximation Error** The combination of off-policy learning, boostrapping, and function approximation is often associated with value overestimation bias (Sutton et al., 2016; Van Hasselt et al., 2018). Prior work has extensively studied such a phenomenon in the context of model-free RL (Anschel et al., 2017; Lan et al., 2020; Moskovitz et al., 2021). Still, off-policy algorithms can be highly sensitive to distribution shifts when data-collection policy diverges far from the target policy and bootstrapping errors compound over time (Kumar et al., 2019). Recent studies in offline RL have taken a huge leap in learning from complete off-policy demonstrations. (Kumar et al., 2020) avoids erroneously overestimating unseen actions by penalizing high-value OOD estimation. In order to avoid extrapolation error, prior work also considered constraint policy toward the behavior policy through weighted behavior cloning (Peng et al., 2019; Fujimoto et al., 2019) or policy regularization (Fujimoto & Gu, 2021; Kumar et al., 2019). Alternatively, Kostrikov et al. (2021); Hansen-Estruch et al. (2023) adopt in-sample learning that do not bootstrap an explicit policy during critic training, thereby avoiding OOD query.

**MPC with Value Prior** Incorporating value prior into the MPC controller reduces its dependency on an imperfect model and has been widely applied in continuous control (Lowrey et al., 2018). Bhardwaj et al. (2020b;a) studies the connection between MPC and entropy regularized RL, enhancing Q-learning accuracy by using model information to calculate value target. Sikchi et al. (2022); Hansen et al. (2022) leverages a value prior learned through model-free RL to approximate the performance of the optimal policy. Building upon this, TD-MPC2 (Hansen et al., 2023) introduces crucial design choices tailored for continuous control tasks, effectively reducing compound model errors and improving learning stability. Due to the decoupled exploration and exploitation, off-policy issues in MBRL can be more severe than model-free algorithms. Hansen et al. (2022); Wang et al. (2025) demonstrate that the nominal policy can be significantly worse than the planner.

## 6 CONCLUSIONS

In this work, we identify a fundamental bottleneck in plan-based model-based reinforcement learning (MBRL): persistent value overestimation arising from a structural policy mismatch between the MPC planner's lookahead policy and the nominal actor. To address this limitation, we proposed TD-M(PC)$^2$, a simple but effective Plan-based MBRL algorithm with a distribution-constrained policy update. The proposed method significantly reduces out-of-distribution queries, curbing overestimation and restoring stable planning performance. Our approach can be seamlessly integrated into the existing framework, requires negligible extra computation, but significantly improves baseline performance, especially on challenging 61-dof humanoid locomotion tasks, demonstrating its practical effectiveness and ease of integration.

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

# A    LLM USAGE DISCLOSURE

We used large language models (ChatGPT and Grammarly APP) *only* for writing assistance (grammar, wording, and minor stylistic edits) on draft text. The model was **not** used for research ideation, data collection, dataset labeling, code generation, experiment design, or analysis. All technical content is authored and verified by human authors. We accept full responsibility for all content in this paper.

# B    THEORY AND PROOF

## B.1    USEFUL LEMMA

**Lemma B.1.** *Singh & Yee (1994) Suppose $\pi_{k+1}$ is 1-step greedy policy of value function $\hat{V}_k$. Denote $V^*$ as the optimal value function, if there exists $\epsilon$ such that $\|V^* - \hat{V}_k\|_\infty \leq \xi_k$, we can bound the value loss of $\pi$ by:*

$$V^* - V^{\pi_{k+1}} \leq \frac{2\gamma\xi_k}{1-\gamma} \tag{12}$$

**Lemma B.2.** *Bertsekas (1996) Suppose $\{\pi_k\}$ is a policy sequence generated by approximate policy iteration (API), then the maximum norm of value loss can be bounded as:*

$$\limsup_{k\to\infty} \|V^* - V^{\pi_k}\|_\infty \leq \frac{2\gamma}{(1-\gamma)^2} \limsup_{k\to\infty} \|\hat{V}_k - V^{\pi_k}\|_\infty \tag{13}$$

**Lemma B.3.** *Sikchi et al. (2022) Denote approximation error for dynamics model $\hat{\rho}$ as $\epsilon_m = \max_{s,a} D_{TV}(\rho(\cdot|s_t, a_t)\|\hat{\rho}(\cdot|s_t, a_t))$. Denote $\epsilon_p$ as the suboptimality incurred in H-step lookahead optimization such that $J^* - \hat{J} \leq \epsilon_p$. Let $\hat{V}$ be an approximate value function such that $\|V^* - \hat{V}\|_\infty \leq \xi$. Also let the reward function $r$ is bounded by $[0, R_{max}]$ and $\hat{V}$ be bounded by $[0, V_{max}]$. Then, the performance of the H-step lookahead policy can be bounded as:*

$$J^{\pi^*} - J^{\pi_H} \leq \frac{2}{1-\gamma^H}\left[C(\epsilon_m, H, \gamma) + \frac{\epsilon_p}{2} + \gamma^H\xi\right] \tag{14}$$

*while $C$ is defined as:*

$$C(\epsilon_m, H, \gamma) = R_{max}\sum_{t=0}^{H-1}\gamma^t t\epsilon_m + \gamma^H H\epsilon_m V_{max} \tag{15}$$

## B.2    PROOF OF THEOREM

**Theorem B.4** (Distribution shift). *Given policies $\pi, \pi' \in \Pi : S \to \Delta(A)$, suppose reward is upper bounded by $R_{max}$, then we have TV-divergence of two visitation distributions lower bounded by the performance gap as:*

$$D_{TV}(p^\pi(s,a)\|p^{\pi'}(s,a)) \geq \frac{1-\gamma}{2R_{max}}|J^\pi - J^{\pi'}| \tag{16}$$

*Proof.* From the definition of expected return, we have the following inequality,

$$\begin{aligned}
|J^\pi - J^{\pi'}| &= |\Sigma_s(p^\pi(s,a) - p^{\pi'}(s,a))r(s,a)| \\
&= |\Sigma_t\Sigma_{s,a}\gamma^t(p^\pi(s,a) - p^{\pi'}(s,a))r(s,a)| \\
&\leq R_{max}\Sigma_t\Sigma_{s,a}\gamma^t|p^\pi(s,a) - p^{\pi'}(s,a)| \\
&= 2R_{max}\Sigma_t\gamma^t D_{TV}(p^\pi(s,a)\|p^{\pi'}(s,a))
\end{aligned}$$

Thus, it is trivial to derive the inequality. ∎

**Corollary B.5** (Policy divergence). *Given policies $\pi, \pi' \in \Pi : S \to \Delta(A)$, suppose reward is upper bounded by $R_{max}$, then we have policy divergence lower bounded by performance gap as:*

$$\max_s D_{TV}\left(\pi'(a|s)\|\pi(a|s)\right) \geq \frac{(1-\gamma)^2}{2R_{max}}|J^\pi - J^{\pi'}| \tag{17}$$

*Proof.* Using Lemma B.1 and Lemma B.2 from Janner et al. (2019), we can relate the visitation divergence to the policy divergence:

$$D_{TV}(p_t^\pi(s,a)\|p_t^{\pi'}(s,a)) \leq D_{TV}(p_t^\pi(s)\|p_t^{\pi'}(s)) + \max_s D_{TV}(p_t^\pi(a|s)\|p_t^{\pi'}(a|s))$$

$$\leq (t+1)\max_s D_{TV}\left(\pi(a|s)\|\pi'(a|s)\right)$$

Given Lemma B.4, we have:

$$|J^\pi - J^{\pi'}| \leq 2R_{max}\Sigma_t\gamma^t(1+t)\max_s D_{TV}\left(\pi(a|s)\|\pi'(a|s)\right)$$

$$\leq \frac{2R_{max}}{(1-\gamma)^2}\max_s D_{TV}\left(\pi(a|s)\|\pi'(a|s)\right)$$

$\square$

**Theorem B.6.** *Suppose $\pi_{k+1}$ is 1-step greedy policy of value function $\hat{V}_k$. Denote $V^*$ as the optimal value function, if there exists $\epsilon$ such that for any $k$, $\|\hat{V}_k - V^{\pi_k}\|_\infty \leq \epsilon$, we can bound the performance of $\pi_{\hat{V}}$ by:*

$$\limsup_{k\to\infty} \|V^* - V^{\pi_{k+1}}\|_\infty \leq \limsup_{k\to\infty} \frac{2\gamma(1+\gamma^2)\epsilon}{(1-\gamma)^3} \tag{18}$$

*Proof.* Directly combining B.1 and B.2.

$$\|V^* - V^{\pi_{k+1}}\|_\infty \leq \frac{2\gamma}{1-\gamma}\|V^* - \hat{V}_k\|_\infty$$

$$\leq \frac{2\gamma}{1-\gamma}\|V^* - \hat{V}_k\|_\infty$$

$$\limsup_{k\to\infty} \|V^* - V^{\pi_{k+1}}\|_\infty \leq \frac{2\gamma}{1-\gamma}\left(\frac{2\gamma}{(1-\gamma)^2} + 1\right)\limsup_{k\to\infty} \|\hat{V}_k - V^{\pi_k}\|_\infty$$

$$\leq \frac{2\gamma(1+\gamma^2)}{(1-\gamma)^3}\epsilon_k$$

$\square$

**Theorem B.7** (Theorem 3.1). *Assume at the k-th iteration, the nominal policy $\pi_k$ is acquired through API and the resulting planner policy is $\pi_{H,k}$. Denote $\epsilon_k$ as the approximation error $\|\hat{V}_k - V^{\pi_k}\|_\infty$ of the learned value function $\hat{V}$. Also denote approximation error (w.r.t. TV-divergence) for dynamics model $\hat{\rho}$ as $\epsilon_m = \max_{s,a} D_{TV}(\rho(\cdot|s_t,a_t)\|\hat{\rho}(\cdot|s_t,a_t))$, planner sub-optimality as $\epsilon_p$. Finally, assume the reward function $r$ is bounded by $[0, R_{max}]$ and $\hat{V}$ is upper bounded by $V_{max}$, then the performance of $\pi_{H,k}$ is bounded w.r.t. the optimal policy as:*

$$\limsup_{k\to\infty} |V^* - V^{\pi_{H,k}}| \leq \limsup_{k\to\infty} \frac{2}{1-\gamma^H}\left[C(\epsilon_{m,k}, H, \gamma) + \frac{\epsilon_{p,k}}{2} + \frac{\gamma^H(1+\gamma^2)}{(1-\gamma)^2}\epsilon_k\right] \tag{19}$$

*while C is defined as:*

$$C(\epsilon_m, H, \gamma) = R_{max}\sum_{t=0}^{H-1}\gamma^t t\epsilon_m + \gamma^H H\epsilon_m V_{max} \tag{20}$$

*Proof.* Denote the planner policy ($H$-step lookahead policy) as $\pi_H$, which is acquired through planning with terminal value $\hat{V}$. We define $\tau^*$ as a trajectory sampled by the optimal policy $\pi^*$, and $\hat{\tau}$ as a trajectory sampled by $\pi_H$ under the true dynamics. We do not consider approximation error for the reward function since knowledge of the reward function can be guaranteed in most training cases. Following the deduction in Theorem 1 of LOOPSikchi et al. (2022), we can have the $H$-step policy suboptimality bound through the following key steps:

$$V^*(s_0) - V^{\pi_{H,k}}(s_0) = \mathbb{E}_{\tau*}\left[\Sigma\gamma^t r(s_t, a_t) + \gamma^H V^*(s_H)\right] - \mathbb{E}_{\hat{\tau}}\left[\Sigma\gamma^t r(s_t, a_t) + \gamma^H \hat{V}_k(s_H)\right]$$

$$\leq \mathbb{E}_{\tau*}\left[\sum \gamma^t r(s_t, a_t) + \gamma^H \hat{V}_k(s_H)\right] - \mathbb{E}_{\hat{\tau}}\left[\sum \gamma^t r(s_t, a_t) + \gamma^H \hat{V}_k(s_H)\right]$$

$$+ \gamma^H \mathbb{E}_{\tau*}\left[V^*(s_H) - \hat{V}_k(s_H)\right] - \gamma^H \mathbb{E}_{\hat{\tau}}\left[V^*(s_H) - \hat{V}_k(s_H)\right]$$

$$+ \gamma^H \mathbb{E}_{\hat{\tau}}\left[V^*(s_H) - V^{\pi_{H,k}}(s_H)\right]$$

$$\leq \frac{2}{1 - \gamma^H}\left[C(\epsilon_{m,k}, H, \gamma) + \frac{\epsilon_{p,k}}{2} + \gamma^H \|V^* - \hat{V}_k\|_\infty\right]$$

Due to the suboptimality, we always have $V^{\pi_H} \leq V^*$. Leveraging the value error bound for API given by Munos (2007), we further bound $\epsilon_v$ with approximation error as:

$$\limsup_{k \to \infty} |V^* - V^{\pi_{H,k}}| \leq \limsup_{k \to \infty} \frac{2}{1 - \gamma^H}\left[C(\epsilon_{m,k}, H, \gamma) + \frac{\epsilon_{p,}}{2} + \gamma^H \|V^* - V^{\pi_k}\|_\infty + \gamma^H \epsilon_k\right]$$

$$\leq \limsup_{k \to \infty} \frac{2}{1 - \gamma^H}\left[C(\epsilon_{m,k}, H, \gamma) + \frac{\epsilon_{p,k}}{2} + \frac{\gamma^H(1 + \gamma^2)}{(1 - \gamma)^2}\epsilon_k\right]$$

This indicates that, given identical conditions, a small approximation error implies that $V^{\pi_H}$ will be close to the optimal value $V^*$. $\qquad\square$

Similar to LOOP, we compare performance bounds between the $H$-step lookahead policy and the 1-step greedy policy (Notice that in API, $\pi_{k+1}$ iteration can be seen as a 1-step greedy policy of $\hat{V}_k$ if we assume the policy improvement step is optimal). With Lemma B.1, we show how value approximation error influences greedy policy performance. Based on Theorem B.6 and Theorem B.7, we conclude that the test-time planning enables $\pi_{H,k}$ to reduce its performance dependency on value accuracy by at least a factor of $\gamma^{H-1}$ compared to the greedy policy $\pi_{k+1}$, providing a general sense of superiority introduce by the planner.

**Theorem B.8** (TD-MPC Error Propagation). *Assume $\pi_{H,k}$ outperforms $\pi_k$ with performance gap $\delta_k = \|V^{\pi_{H,k}} - V^{\pi_k}\|_\infty$. Denote value approximation error $\epsilon_k = \|\hat{V}_k - V^{\pi_k}\|_\infty$, approximated dynamics holds model error $\epsilon_{m,k}$, planner sub-optimality is $\epsilon_{p,k}$. Also let the reward function $r$ is bounded by $[0, R_{max}]$ and $\hat{V}_{k-1}$ be bounded by $[0, V_{max}]$, then the following uniform bound of performance gap holds:*

$$\delta_k \leq \frac{1}{1 - \gamma^H}\left[2C(\epsilon_{m,k-1}, H, \gamma) + \epsilon_{p,k-1} + (1 + \gamma^H)\delta_{k-1} + \frac{2\gamma(1 + \gamma^{H-1})}{1 - \gamma}\epsilon_{k-1}\right] \qquad (21)$$

*where $C$ is defined as equation 15.*

*Proof.* Denote the planner policy ($H$-step lookahead policy) as $\pi_H$, which is acquired through planning with terminal value $\hat{V}$. We do not consider approximation error for the reward function since knowledge of the reward function can be guaranteed in most training cases. At k-th iteration, denote $\hat{\tau}^k$ as the trajectory sampled by the planner policy; $\tau^k$ as trajectory sampled by optimal planner leveraging real dynamics model; $\tau^{\pi_k}$ as a trajectory sampled by nominal policy $\pi_k$ (all trajectories are sampled in the environment rather than under approximate dynamics model). For simplicity, $\Sigma$ in this proof stands for $\Sigma_{t=0}^{H-1}$ if not specified.

$$V^{\pi_{H,k}}(s_0) - V^{\pi_k}(s_0) = \mathbb{E}_{\hat{\tau}^k}\left[\Sigma\gamma^t r(s_t, a_t) + \gamma^H V^{\pi_{H,k}}(s_H)\right] - V^{\pi_k}(s_0)$$

$$= \mathbb{E}_{\hat{\tau}^k}\left[\Sigma\gamma^t r(s_t, a_t) + \gamma^H V^{\pi_k}(s_H)\right] - V^{\pi_k}(s_0)$$

$$+ \gamma^H \mathbb{E}_{\hat{\tau}^k}\left[V^{\pi_{H,k}}(s_H) - V^{\pi_k}(s_H)\right]$$

Particularly, we have:

$$\mathbb{E}_{\hat{\tau}^k}[\Sigma\gamma^t r(s_t, a_t) + \gamma^H V^{\pi_k}(s_H)] - V^{\pi_k}(s_0)$$

$$= \mathbb{E}_{\hat{\tau}^k}[\Sigma\gamma^t r(s_t, a_t) + \gamma^H \hat{V}_{k-1}(s_H)] + \gamma^H \mathbb{E}_{\hat{\tau}^k}[V^{\pi_k}(s_H) - \hat{V}_{k-1}(s_H)] - V^{\pi_k}(s_0)$$

$$\leq \mathbb{E}_{\tau^{k-1}}[\Sigma\gamma^t r(s_t, a_t) + \gamma^H \hat{V}_{k-1}(s_H)] - \mathbb{E}_{\hat{\tau}^{k-1}}[\Sigma\gamma^t r(s_t, a_t) + \gamma^H \hat{V}_{k-1}(s_H)]$$

$$+ \mathbb{E}_{\hat{\tau}^{k-1}}[\Sigma\gamma^t r(s_t, a_t) + \gamma^H \hat{V}_{k-1}(s_H)] + \gamma^H \mathbb{E}_{\hat{\tau}^k}[V^{\pi_k}(s_H) - \hat{V}_{k-1}(s_H)] - V^{\pi_k}(s_0)$$

$$\leq \mathbb{E}_{\tau^{k-1}}[\Sigma\gamma^t r(s_t, a_t) + \gamma^H \hat{V}_{k-1}(s_H)] - \mathbb{E}_{\hat{\tau}^{k-1}}[\Sigma\gamma^t r(s_t, a_t) + \gamma^H \hat{V}_{k-1}(s_H)]$$

$$+ \mathbb{E}_{\hat{\tau}^{k-1}}[\Sigma\gamma^t r(s_t, a_t) + \gamma^H \hat{V}_{k-1}(s_H)] + \gamma^H \mathbb{E}_{\hat{\tau}^k}[V^{\pi_k}(s_H) - \hat{V}_{k-1}(s_H)]$$

$$- V^{\pi_{k-1}}(s_0) + [V^{\pi_{k-1}}(s_0) - V^{\pi_k}(s_0)]$$

The second step is due to the definition of $\tau^{k-1}$:

$$\tau^{k-1} = \arg\max_{\tau} \mathbb{E}_{\tau}[\Sigma\gamma^t r(s_t, a_t) + \gamma^H \hat{V}_{k-1}(s_H)]$$

We use Theorem 1 in Sikchi et al. (2022) for the first row to bind them by model error and planner sub-optimality.

$$\mathbb{E}_{\tau^{k-1}}[\Sigma\gamma^t r(s_t, a_t) + \gamma^H \hat{V}_{k-1}(s_H)] - \mathbb{E}_{\hat{\tau}^{k-1}}[\Sigma\gamma^t r(s_t, a_t) + \gamma^H \hat{V}_{k-1}(s_H)] \leq 2C(\epsilon_{m,k-1}, H, \gamma) + \epsilon_{p,k-1}$$

where $C(\epsilon_{m,k-1}, H, \gamma)$ is the same as defined in Theorem B.7.

For the second row, since we want to construct a bound related to $\|V^{\pi_{H,k-1}} - V^{\pi_{k-1}}\|$. Under this orientation, we first show that:

$$\mathbb{E}_{\hat{\tau}^{k-1}}[\Sigma\gamma^t r(s_t, a_t) + \gamma^H \hat{V}_{k-1}(s_H)] + \gamma^H \mathbb{E}_{\hat{\tau}^k}[V^{\pi_k}(s_H) - \hat{V}_{k-1}(s_H)]$$

$$= \mathbb{E}_{\hat{\tau}^{k-1}}[\Sigma\gamma^t r(s_t, a_t) + \gamma^H V^{\pi_{H,k-1}}(s_H)] + \gamma^H \mathbb{E}_{\hat{\tau}^{k-1}}[\hat{V}_{k-1}(s_H) - V^{\pi_{H,k-1}}(s_H)]$$

$$+ \gamma^H \mathbb{E}_{\hat{\tau}^k}[V^{\pi_k}(s_H) - V^{\pi_{k-1}}(s_H)] + \gamma^H \mathbb{E}_{\hat{\tau}^k}[V^{\pi_{k-1}}(s_H) + \hat{V}_{k-1}(s_H)]$$

$$= V^{\pi_{H,k-1}}(s_0) + \gamma^H \mathbb{E}_{\hat{\tau}^{k-1}}[\hat{V}_{k-1}(s_H) - V^{\pi_{k-1}}(s_H)] + \gamma^H \mathbb{E}_{\hat{\tau}^{k-1}}[V^{\pi_{k-1}}(s_H) - V^{\pi_{H,k-1}}(s_H)]$$

$$+ \gamma^H \mathbb{E}_{\hat{\tau}^k}[V^{\pi_k}(s_H) - V^{\pi_{k-1}}(s_H)] + \gamma^H \mathbb{E}_{\hat{\tau}^k}[V^{\pi_{k-1}}(s_H) - \hat{V}_{k-1}(s_H)]$$

Sequentially, we combine the inequalities above:

$$\mathbb{E}_{\hat{\tau}^k}[\Sigma\gamma^t r(s_t, a_t) + \gamma^H V^{\pi_k}(s_H)] - V^{\pi_k}(s_0)$$

$$\leq 2C(\epsilon_{m,k-1}, H, \gamma) + \epsilon_{p,k-1} + 2\gamma^H \epsilon_{k-1}$$

$$+ V^{\pi_{H,k-1}}(s_0) - V^{\pi_{k-1}}(s_0) + \gamma^H \mathbb{E}_{\hat{\tau}^{k-1}}[V^{\pi_{k-1}}(s_H) - V^{\pi_{H,k-1}}(s_H)]$$

$$+ \gamma^H \mathbb{E}_{\hat{\tau}^k}[V^{\pi_k}(s_H) - V^{\pi_{k-1}}(s_H)] + [V^{\pi_{k-1}}(s_0) - V^{\pi_k}(s_0)]$$

We can have a rough bound on the second line by $(1 + \gamma^H)\delta_{k-1}$. With Lemma 6.1 in Bertsekas (1996), we can bound the final line as:

$$\gamma^H \mathbb{E}_{\hat{\tau}^k}[V^{\pi_k}(s_H) - V^{\pi_{k-1}}(s_H)] + [V^{\pi_{k-1}}(s_0) - V^{\pi_k}(s_0)] \leq (1 + \gamma^H)\frac{2\gamma}{1-\gamma}\epsilon_{k-1}$$

Combining all the inequalities and leveraging the contraction property, we get the final bound for the performance gap:

$$V^{\pi_{H,k}}(s_0) - V^{\pi_k}(s_0) \leq \frac{1}{1-\gamma^H}\left[2C(\epsilon_{m,k-1}, H, \gamma) + \epsilon_{p,k-1} + (1 + \gamma^H)\delta_{k-1} + \frac{2\gamma(1 + \gamma^{H-1})}{1-\gamma}\epsilon_{k-1}\right]$$

Thus, we can easily arrive at the final result. $\qquad\square$

## C    EXTENSIVE RELATED WORKS

**Model-based RL.**    The core of model-based reinforcement learning is how to leverage the world model to recover a performant policy. Recent advances aim to balance scalability and adaptability by integrating strengths from both model-free and model-based paradigms. Dyna-Q Sutton (1991) considers using simulated rollouts from a learned model to augment real-world experience, often referred to as "background planning" Hamrick et al. (2020). Janner et al. (2019) provides a theoretical guarantee of monotonic policy improvement with model-generated data. Katsigiannis & Ramzan (2017); Hafner et al. (2020; 2023) further introduce latent world models for high-dimensional tasks, e.g., visual control. Planning algorithms such as Model Predictive Control (MPC) and Monte Carlo Tree Search (MCTS) can explicitly exploit model knowledge to acquire a superior control policy. Planner is critical to policy learning, it constantly interacts with other components through high-quality data acquisition Lowrey et al. (2018); Hansen et al. (2022), inference-time planning Chua et al. (2018); Hafner et al. (2019), or provides expert learning signal Schrittwieser et al. (2020); Bhardwaj et al. (2020b); Ye et al. (2021); Wang et al. (2024). Schrittwieser et al. (2020) achieves trust-region policy optimization through combining MCTS with policy and value prior Grill et al. (2020). Following this line of work, Ye et al. (2021); Wang et al. (2024; 2025) leverages expert iteration to provide up-to-date planner policy and value estimation as a learning target. Contrary to these methods, we do not rely on expert iteration for policy improvement, but leverage past behavior policy to enforce conservatism.

**Off-policy Learning and Value Approximation Error**    The combination of off-policy learning, boostrapping, and function approximation is often associated with value overestimation bias Sutton et al. (2016); Van Hasselt et al. (2018) and has been a long-standing problem in DRL. Prior work has extensively studied such a phenomenon in the context of model-free RL Anschel et al. (2017); Lan et al. (2020); Moskovitz et al. (2021). Still, off-policy algorithms can be highly sensitive to distributional shifts when data-collection policy diverges far from target policy, leading to instability and poor generalization as bootstrapping errors compound over time Kumar et al. (2019). Extrapolation errors have been well articulated in offline RL studies Kumar et al. (2019); Peng et al. (2019); Levine et al. (2020), addressing this issue enables algorithms to learn from complete off-policy demonstrations. Such an issue appears when the value function is queried with *out-of-distribution* (OOD) state-action pairs. Then, temporal difference methods propagate generalization errors iteratively, causing the value estimation to deviate further. Many methods are proposed to resolve this by addressing conservatism, such as in-distribution learning Kostrikov et al. (2021); Garg et al. (2023), conservative evaluation Kumar et al. (2020), weighted imitation Peng et al. (2019); Nair et al. (2020); Hansen-Estruch et al. (2023), or behavior regularization Fujimoto & Gu (2021); Lu et al. (2023).

While the training dataset is fixed in offline RL, exploration is critical in online RL. For model-free algorithms where exploration and exploitation policies align closely, with stabilizing methods applied to value learning Fujimoto et al. (2018); Anschel et al. (2017), such value overestimation bias is expected to be fixed since the corresponding policy tends to visit these overestimated regions. And in cases such that the remedy to value learning is conservative enough, e.g., taking the minimum value of an ensemble of value functions, even underestimation could take place Hasselt (2010); Lan et al. (2020). However, as demonstrated in 3.2, the usage of online planning distinguishes it from the issue encountered in model-free off-policy learning.

**MPC with Value Prior**    Incorporating value prior into the MPC controller reduces its dependency on an imperfect model. POLO Lowrey et al. (2018) uses a value prior as the terminal cost function in an MPPI planner to efficiently search the trajectory space; it also maintains an ensemble of value functions to track uncertainties and promote optimistic exploration. Bhardwaj et al. (2020b;a) studies the connection between MPC and entropy regularized RL, enhancing Q-learning accuracy by using model information to calculate the value target. Sikchi et al. (2022); Hansen et al. (2022) leverages a value prior learned through model-free RL to approximate the performance of the optimal policy. Building upon this, TD-MPC2 Hansen et al. (2023) introduces crucial design choices tailored for continuous control tasks, effectively reducing compound model errors and improving learning stability. Due to the decoupled exploration and exploitation, off-policy issues in policy-based MBRL can be more severe than model-free algorithms. Hansen et al. (2022); Wang et al. (2025) observed that the nominal policy is significantly worse than the planner policy in perfor-

---

**Algorithm 1** TD-M(PC)$^2$

---

**Require:** $\theta, \psi, enc, \phi, \phi', P, \alpha, \beta, \rho, \gamma$

  Initialize policy network $\pi_\theta$, latent world model $d_\psi$, encoder $enc$, and value functions $Q_\phi, Q_{\phi'}$ by pertaining on uniformly sampled data

  **for** each training step **do**

    **if** collect data **then**

      Planning by Algorithm 2: $a \sim \mu = P(enc(s), \pi_\theta, Q_\phi, d, \gamma)$.

      Environment step: $r, s', done = env.step(a)$

      Add $(s, a, \mu, r, s')$ to buffer $\mathcal{B}$.

    **end if**

    Sample trajectories $\{(s_t, a_t, \mu_t, r_t, s_{t+1})_{0:H}\} \sim \mathcal{B}$

    **# Model Update**

    Calculate TD target by bootstrapping $\pi_\theta$

    Update $d_\psi, enc, Q_\phi$ by equation 7

    **# Constrained Policy Update**

    Calculate policy loss by equation 10 and update $\theta$

    **Polyak update** $\phi'_i = \rho\phi'_i + (1 - \rho)\phi_i,\ i = 1, 2$

  **end for**

  **return** $\theta, \phi, P$

---

**Algorithm 2** MPPI

---

**Require:** $z_0, \pi, Q(z, a), d, \lambda, \gamma, N, N_\pi, H$

  Initialize $\mu^0$ as concat$\{p[:-1], \mathbf{0}\}$, initialize $\sigma^0$

  **for** iteration i=0, 1, ..., I **do**

    Sample $N$ action traj. from len. $H$ from $\mathcal{N}(\mu_i; \sigma_i^2 \mathbf{I})$

    Sample $N_\pi$ action traj. by rollout $\pi$ in latent dynamics $d$

    Collect all trajectories $\tau$

    **# Rollout and estimate discounted returns**

    **for** All traj. $j = 1, \ldots, N + N_\pi$ **do**

      $R_j = 0$

      **for** step t=0, 1, ..., H-1 **do**

        $z_{t+1}, r_t = d(z_t, a_t)$

        $R_j = R_j + \gamma^t r_t$

      **end for**

      $R_j = R_j + \gamma^H Q(z_H, \pi(z_H))$

    **end for**

    **# Update $\mu$ and $\sigma$**

    Select top-K trajectories $\{\tau_1, \ldots, \tau_K\}$

    Calculate score $\omega_k = \frac{\exp(\lambda R_k)}{\Sigma_{i=1}^{K} \exp(\lambda R_k)}$

    Update parameters $\mu^{i+1} = \Sigma_{i=1}^{K} \omega_k \tau_k,\ \ \sigma^{i+1} = \sqrt{\Sigma_{i=1}^{K} \omega_k (\tau_k - \mu^{i+1})^2}$

  **end for**

  **return** $\mathcal{N}(\mu^I; (\sigma^I)^2 \mathbf{I})$

---

mance, Wang et al. (2025) attributes this to the low efficiency of policy learning. However, we argue that while exploitation is surely Sikchi et al. (2022); Argenson & Dulac-Arnold (2020) tackles this policy divergence by mixing action sequences proposed by the policy prior with randomly sampled trajectories into MPPI. In comparison, our method effectively addresses overestimation without compromising the planner.

## D  DESIGN CHOICES AND IMPLEMENTATION DETAILS

### D.1  ALGORITHM FORMULATION

To further elaborate on the design choices, we present the general formulation of constrained policy iteration, followed by a detailed discussion of its implementation and interaction with other critical components within TD-MPC.

Given the behavior policy $\mu(\cdot|s) = \Sigma_{k=0}^K \omega_k \pi_{H,k}(\cdot|s)$ (following the notation in Peng et al. (2019)), Conservative policy improvement step can be interpreted as finding a solution in the trust region near $\mu$. The optimal solution of problem equation 8 is the combination of behavior policy and Boltzmann distribution, with partition function $Z(s)$ and Lagrangian multiplier $\beta$:

$$\mu^*(a|s) := \frac{1}{Z(s)}\mu(a|s)\exp(\frac{1}{\beta}Q(s,a)), \quad Z(s) := \int_a \mu(a|s)\exp(\frac{1}{\beta}Q(s,a)) \tag{22}$$

In principle, the training objective can be formulated using either the reverse KL-divergence (RKL) $D_{KL}(\pi\|\mu^*)$, or forward KL-divergence (FKL) $D_{KL}(\mu^*\|\pi)$. For online off-policy setting, beyond its well-known "zero-avoiding" behavior, prior studies Chan et al. (2022) have shown that FKL encourages mode-covering but does not guarantee policy improvement, often leading to degraded performance, especially under large entropy regularization. In E, we show that FKL could mitigate the value overestimation problem for high-dimensional tasks but may lead to training instability.

Based on these observations, we choose RKL style policy learning Fujimoto & Gu (2021). Instead of directly calculating the log-likelihood of $\mu$, we maximize $\mathbb{E}_{\mu'\sim\mathcal{B}}[\log\mu']$ as its lower bound. Such a surrogate greatly simplifies the calculation. Specifically, the policy improvement step can be realized as follows to leverage a sequence of transitions:

$$\pi_{k+1} \leftarrow \underset{\pi}{\arg\min}\mathbb{E}_{\{s,a',\mu\}_{0:H-1}\sim\mathcal{B}}\Big[\sum_{t=0}^H \lambda^t \mathbb{E}_{a_t\sim\pi(\cdot|z_t)}\big[-Q^{\pi_k}(z_t,a_t)/S_q + \alpha\log(\pi(a_t|z_t))$$
$$- \beta \cdot \frac{d_{\text{action}}}{d_{\max}}\log(\mu_t(a_t))/S_q\big]\Big], \tag{23}$$
$$z_0 = h(s_0), \quad z_{t+1} = d(z_t, a_t')$$

Notably, we denote $a'_{0:H-1}$ as the behavior action sampled from the buffer. Thus, unlike dreamer Katsigiannis & Ramzan (2017), we do not actually roll out $\pi$ and leverage simulated experience during training. We also scale the training loss by a moving percentile $S_q \leq (Q_{\max} - Q_{\min})$ of the value function to improve stability.

$$S_q = \text{EMA}\left(\max\{\text{Per}(Q,95) - \text{Per}(Q,5), 1\}, \xi\right) \tag{24}$$

**Conservative Threshold** $s$    We notice that overly addressing policy constraints during the initial stage sometimes results in failure to escape from the local minima. Thus, we maintain a moving percentile $S_q$ for the Q function as in Hansen et al. (2022; 2023) and leverage an adaptive curriculum on $\beta$:

$$\beta = \begin{cases} 0, & \text{if } S_q < s \\ \beta, & \text{otherwise} \end{cases}$$

The intuition behind this scheduler is that a small $S_q$ could be an indicator of the initial training phase. Also, decreasing $S_q$ indicates a stable or non-improving performance where exploration is required. While one may argue that other deliberately designed curricula could be more effective, we find that this straightforward setting is sufficient. For exploration-intensive tasks like `humanoid-run` in DMControl, this curriculum allows approximately 100k environment steps at the beginning without constraints enforced on the nominal policy. Empirically, the "free-explore" stage could incur a certain degree of value overestimation, but it will soon be rectified once the conservative term takes effect. For most tasks with denser reward signals, its impact on performance is minimal. But carefully tuned $s$ yields better performance Section E.

**Conservative Coefficient**    We found scale $\beta$ properly with action dimensionality, as the sparsity of high-dimensional space leads to a more severe distribution shift. We suggest $\beta \propto \frac{d_{\text{action}}}{d_{\max}}$ is a good choice that balances performance on both low- and high-dimensional tasks. The reported result in the main paper applied this scaling, as the conservatism coefficient is technically $\beta\frac{d_{\text{action}}}{d_{\max}}$.

We have the same training objective as TD-MPC2 when it comes to the dynamics model, reward model, value function, and encoder:

$$\mathcal{L} = \mathop{\mathbb{E}}_{(s,a,r,s')_{0:H}} \Big[ \sum_{t=0}^{H} \gamma^t \Big( c_d \cdot \|d(z,a,e) - sg(h(s'_t))\|_2^2 + c_r \cdot CE(\hat{r}_t, r_t) + c_q \cdot CE(\hat{q}_t, q_t) \Big) \Big] \tag{25}$$

Where the reward function and value function's output are discretized and updated with cross-entropy loss given their targets.

In addition, we disclose implementation distinctions for baseline variants used in section 4.3. We directly update the policy for the behavior cloning version ($\beta = \infty$) by maximizing the log-likelihood term. Following Hansen et al. (2022; 2023), we introduce a moving percentile $S$ to scale the magnitude of the loss:

$$\mathcal{L}_\pi = \mathbb{E}_{s\sim\mathcal{B}}\mathbb{E}_{a\sim\pi(\cdot|s)} \log \mu(a \mid s) / \max(1, S) \tag{26}$$

### D.2 IMPLEMENTATION DETAILS

Table 1 lists the full hyperparameter settings for our training procedure, planner, actor (nominal policy), critic, and network architectures. To prevent the planner's rollout distribution from collapsing to a deterministic policy—and to improve numerical stability—we constrain its action-noise standard deviation to lie within $[\texttt{Min\_Planner\_Std}, \texttt{Max\_Planner\_Std}]$. Importantly, the planner in Algorithm 2 combines Model Predictive Path Integral control (MPPI) (Williams et al., 2016) with the Cross-Entropy Method (CEM) (Rubinstein & Kroese, 2004), using a top-$K$ selection mechanism: 5% of trajectories follow the nominal policy $\pi$, while the remainder are derived from planner sampling, allows the planner to diverge far from the nominal policy $\pi$.

All model components—dynamics and reward model, encoder, actor, and critic—are implemented as three-layer MLPs with Mish activations and LayerNorm. Consistent with TD-MPC2 Hansen et al. (2023), we apply SimNorm to the latent state representations. We optimize all networks using Adam (Kingma & Ba, 2014) with gradient clipping. Experiments were run on a single NVIDIA RTX A6000 GPU paired with an AMD EPYC 7513 32-Core CPU. For 2 M environment steps and UTD=1, each training session completes in approximately 38.5 hours, with a minimal fluctuation across environments. The proposed algorithm does require more storage compared to TD-MPC2, but we consider that insignificant. Since we store the mean and std of the planner policy, whose additional budget is of the same scale as the action storage and smaller than the observation storage. For a replay buffer of 1M steps and state-based observation, the extra memory requirement is 0.01 GB (from 0.03 to 0.04GB) for *DMControl* tasks and 0.6 GB (from 0.8 to 1.4GB) for *HumanoidBench*, which is far from a heavy burden.

## E DISCUSSIONS AND ADDITIONAL RESULTS

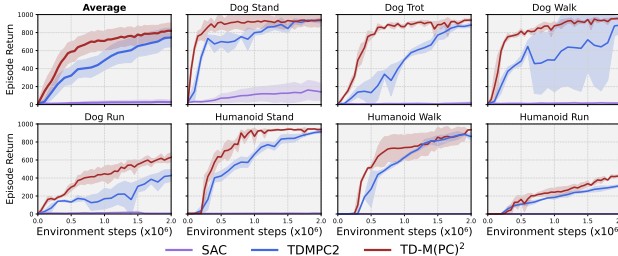

Figure 5: **Performance on DMControl suite (7 High-dimensional tasks).** Average episode return of our method and baselines. We report mean evaluation performance and 95% CIs across 7 high-dimensional continuous control tasks.

Besides the training curves on 23 DMControl tasks (Figure 6 and Figure 5), we provide further discussions and empirical results on value approximation error, different methods to enforce conservatism, and additional ablations.

Table 1: Hyperparameter settings. We directly apply settings in Hansen et al. (2023) for the shared hyperparameters without further tuning. We share the same setting across all tasks demonstrated before.

| Hyperparameter | Value |
|---|---|
| **Training** | |
| Learning rate | $3 \times 10^{-4}$ |
| Batch size | 256 |
| Buffer size | 1_000_000 |
| Sampling | Uniform |
| Reward loss coefficient ($c_r$) | 0.1 |
| Value loss coefficient ($c_q$) | 0.1 |
| Consistency loss coefficient ($c_d$) | 20 |
| Discount factor ($\gamma$) | 0.99 |
| Target network update rate | 0.5 |
| Gradient Clipping Norm | 20 |
| Optimizer | Adam |
| Up-to-data (UTD) | 1 |
| **Planner** | |
| MPPI Iterations | 6 |
| Number of samples | 512 |
| Number of elites | 64 |
| Number policy rollouts | 24 |
| horizon | 3 |
| Minimum planner std | 0.05 |
| Maximum planner std | 2 |
| Temperature ($\lambda$) | 0.5 |
| **Actor** | |
| Minimum policy log std | -10 |
| Maximum policy log std | 2 |
| Entropy coefficient ($\alpha$) | $1 \times 10^{-4}$ |
| **Prior constraint coefficient** ($\beta$) | 1.0 |
| Scale Threshold ($s$) | 2.0 |
| **Critic** | |
| Q functions Esemble | 5 |
| Number of bins | 101 |
| Minimum value | -10 |
| Maximum value | 10 |
| **Architecture(5M)** | |
| Encoder layers | 2 |
| Encoder dimension | 256 |
| Hidden layer dimension | 512 |
| Latent space dimension | 512 |
| Task embedding dimension | 96 |
| Q function drop out | 0.01 |
| Activation | Mish |
| Normalization | LayerNorm |
| SimNorm dimension | 8 |

**Value Approximation Error** In Section 3.1, we illustrated value overestimation by comparing the true value with the value function's estimate. The true value is approximated using Monte Carlo sampling as $\frac{1}{N}\Sigma_{n=1}^{N}[R(\tau_n^\pi)]$, where $\tau_n^\pi$ is trajectory following the nominal policy $\pi$. Unlike the approach to demonstrate overestimation in Fujimoto et al. (2018) that averages over states drawn i.i.d. from the buffer, we sample all trajectories starting from the initial state $s_0 \sim \rho_0$. Accordingly, the function estimation is calculated by averaging the action value following $\pi$ at the initial state as $\mathbb{E}_{s\sim\rho_0,a\sim\pi(\cdot|s)}[\hat{Q}(s,a)]$. We argue that this approach more effectively illustrates the overestimation

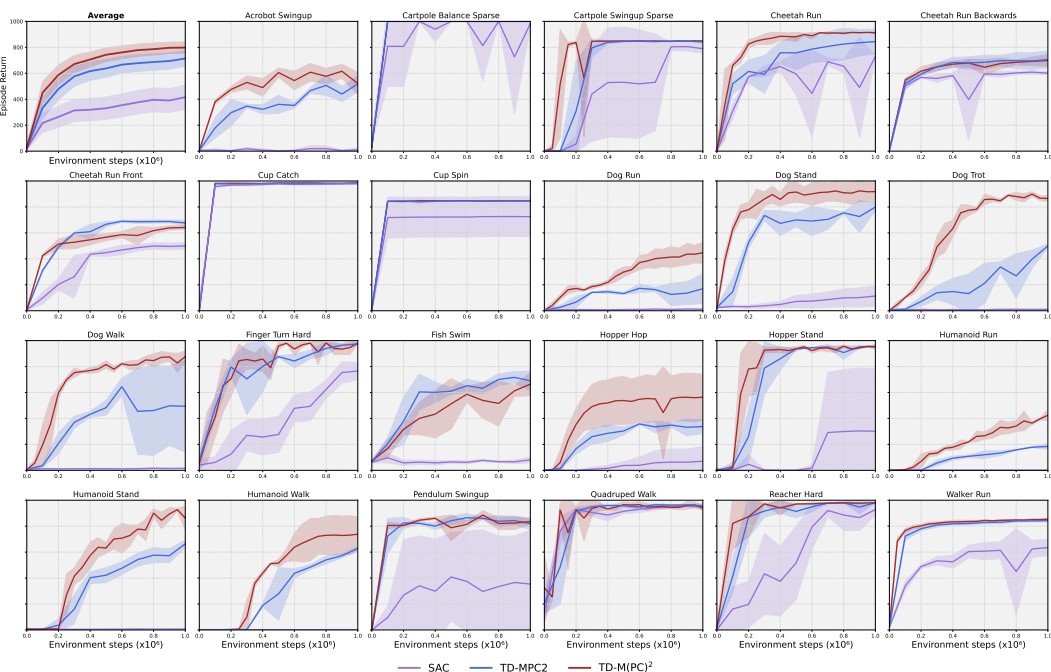

Figure 6: **Performance on DMControl suite.** Average episode return of our method and baselines. We report mean evaluation performance and 95% CIs across 16 low- or medium-dimensional and 7 high-dimensional continuous control tasks within **1M** environment steps.

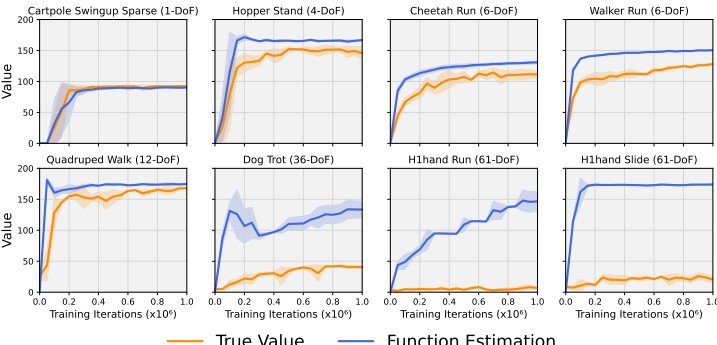

Figure 7: **TD-MPC2 value estimation bias.** We visualize value overestimation across a diverse range of tasks from *DMControl* or *Humanoidbench*. Overestimation bias scales with the action dimension of the environment predictably.

phenomenon, since value approximation errors propagate through TD learning and accumulate at the initial state Sutton & Barto (2018), making overestimation more pronounced and easier to observe. Additional evaluations on the value overestimation bias on low- and medium-dimensional tasks can be found in Figure 7.

**Planning horizon** We compare the approximation error between TD-MPC2 with a planning horizon of 1 and a horizon of 3 using `h1hand-run-v0` task. As shown in Figure 10, while both versions exhibit significant overestimation bias, the patterns of error growth differ. Over 2M training steps, the error in the horizon-1 version grows nearly linearly, showing no clear trend of convergence. In contrast, although the horizon-3 version initially accumulates errors more rapidly, its error growth rate gradually decreases over time.

We further justify our theoretical results through an ablation study on the planning horizon. By combining Theorem B.1 and Theorem B.3, we know that $\pi_H$'s dependency on value error is scaled by a factor of $\frac{\gamma^{H-1}(1-\gamma)}{(1-\gamma^H)}$ relative to the greedy policy's dependency on value error. Consequently, given the same $\hat{V}$, we expect the performance gap between the H-step lookahead policy $\pi_{H,k}$ and the greedy policy $\pi_{k+1}$ to be smaller in the early stages of training, which aligns with the lower approximation error initially observed. However, according to Theorem 3.2, shorter horizons amplify the error accumulation term, resulting in a faster growth rate. Therefore, this empirical observation further supports our theoretical analysis.

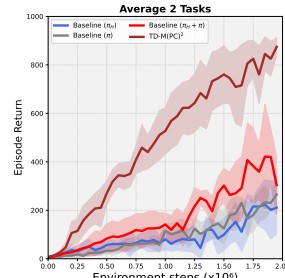

Figure 8: **Exploration policy formulation.** Average performance on two high-dimensional humanoid tasks. Report mean and 95% CIs across 3 random seeds.

**Exploration policy** In addition to the discussion in Section 3.2, we compare the performance of TD-MPC2 with different exploration strategy in Figure 8: a) Baseline ($\pi_H$): vanilla TD-MPC2, solely leveraging planner policy $\pi_H$ for exploration b) Baseline ($\pi$): TD-MPC2 solely leveraging nominal policy $\pi$ for exploration, equivalent to SAC at training, only leverage planner for test-time inference; c) Baseline ($\pi_H + \pi$): TD-MPC2 with 50% trajectories collected through the nominal policy $\pi$;

**Conservative Threshold** We consider evaluating different choices of $s$. Empirically, the algorithm is not sensitive to $s$ for most cases except for *humanoid-run*. As shown in Section E, the larger $s$ performs better. This might be due to the task's spare reward signal in the initial stage. It requires a better exploration to get the humanoid up in the first place. Still, the main results are evaluated with a fixed $s = 2$. Results are reported over 3 seeds at **1M** environment steps.

Table 2: Ablation on conservative threshold $s$

| **Task** | dog-trot | humanoid-run | h1hand-slide | h1hand-run |
|---|---|---|---|---|
| $s = 2$ | 866.2±19.7 | 241.9±38.5 | 492.9±93.1 | 503.9±231.1 |
| $s = 10$ | 831.2±64.1 | 424.7±35.0 | 435.3±41.8 | 228.2±38.2 |

**Conservative Policy Learning** Offline RL algorithms aim to stabilize the learning process and improve policy performance by carefully handling unseen data. However, we argue that not all offline RL methods are well-suited for the TD-MPC setting. We empirically found that the FKL algorithm performs worse than the bc-constrained RKL policy learning despite the theoretical equivalence. In Figure 11, *AWAC-MPC* refers to the variant that employs AWAC Nair et al. (2020) for constrained policy iteration.

Its implementation is based on CORL. These findings are aligned with Park et al. (2024), which demonstrates the advantage of bc-constrained policy learning over AWR/AWAC due to the encouragement of mode-seeking.

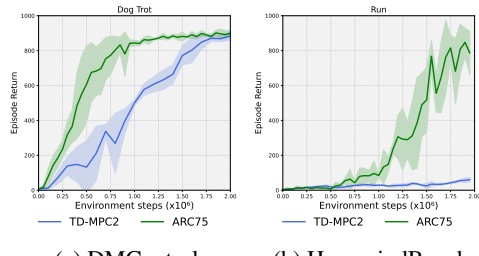

(a) DMContorl     (b) HumaniodBench

Figure 9: **Constraining the planner.** Increasing the percentage of trajectories proposed by the policy prior improves data efficiency in high-dimensional tasks.

Moreover, we do not recommend employing conservative Q-learning methods Kumar et al. (2020). Such methods are designed to penalize the Q-values of out-of-distribution (OOD) actions, ensuring that the agent remains within the boundaries of the training data. While this helps to prevent

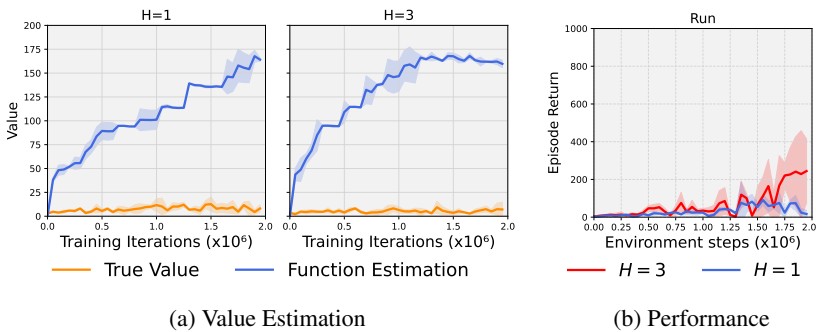

(a) Value Estimation                    (b) Performance

Figure 10: TD-MPC2 ablation results of horizon at `h1hand-run-v0`. (a) Value estimation error with different planning horizons; (b) Episode return with different horizons. The variant with a longer horizon shows a convergent error growth pattern.

overestimation, it may introduce a significant drawback: a consistent underestimation of the overall Q-value function. This underestimation not only affects out-of-distribution data but also reduces the scale of the Q-values overall Nakamoto et al. (2024). As a result, value-guided planning becomes excessively cautious, disincentivizing the selection of novel actions outside the buffer. This overly conservative behavior severely limits the agent's ability to explore, which, however, is a key aspect of online reinforcement learning. We favor TD3-BCFujimoto & Gu (2021) or BC-SACLu et al. (2023) style algorithm for this particular problem setting.

**Regularizing the Planner**  Since policy mismatch between the online planning and the nominal policy (actor) leads to value approximation error, one straightforward solution is to ensure the planner policy stays close to the actor. LOOP Sikchi et al. (2022) proposes Actuator Regularized Control (ARC), which modifies the original MPPI sampling by mixing a portion of actor-proposed trajectories. For the TD-MPC pipeline, a small percentage of mixture no longer ensures a shifted distribution due to the top-K mechanism. To study the effectiveness of this "forward regularization", we increased this ratio from 5% to 75%, while other components remain the same as TD-MPC2. This variant is referred to as "ARC75". As shown in Figure 9, this method also significantly boosts performance on high-dimensional control tasks.

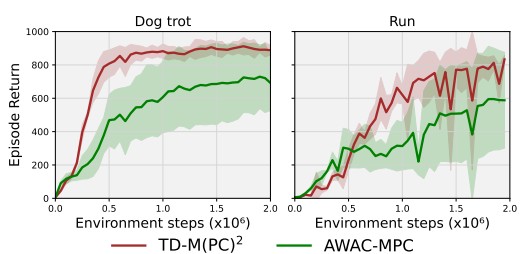

Figure 11: **Constrained policy update through AWAC.** Mean and 95% CIs across 3 random seeds on high-dimensional tasks.

**Model Bias**  In addition to direct OOD query, model biasKumar et al. (2019) is also considered an essential source of extrapolation error for offline RL: Due to a limited number of transitions contained in the training dataset, TD-target does not strictly reflect an estimation of real transitions. For an online off-policy problem, the problem is not critical since the buffer is continuously updated.

## F  LIMITATIONS AND FUTURE WORKS

This paper primarily focuses on single-task state-space RL problems. It would be interesting to evaluate the generalizability of our method on visual RL and multi-task RL settings. Moreover, we are interested in further studying our method under deployment-efficient settings Matsushima et al. (2020) to facilitate real-world applications.

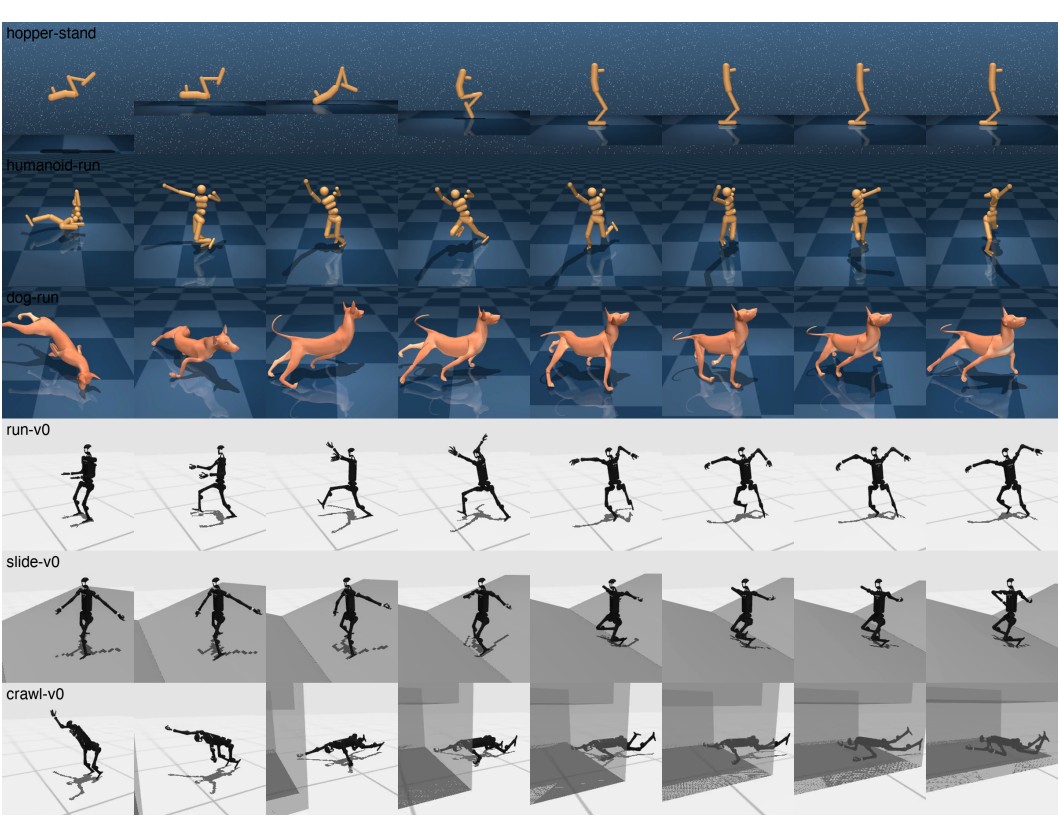

Figure 12: **Task Visualization.** We demonstrate trajectories generated by our method on 6 tasks across two benchmarks (DMControl and HumanoidBench) as qualitative results; tasks are listed as follows: `Hopper-stand` ($\mathcal{A} \in \mathbb{R}^4$), `Humanoid-run` ($\mathcal{A} \in \mathbb{R}^{21}$), `Dog-run` ($\mathcal{A} \in \mathbb{R}^{36}$), `h1hand-run-v0` ($\mathcal{A} \in \mathbb{R}^{61}$), `h1hand-slide-v0` ($\mathcal{A} \in \mathbb{R}^{61}$), `h1hand-crawl-v0` ($\mathcal{A} \in \mathbb{R}^{61}$).

