# OpenReview forum: "TD-M(PC)$^2$: Improving Temporal Difference MPC Through Policy Constraint"
_ICLR.cc/2026/Conference — ICLR 2026 Conference Withdrawn Submission_

### Official Review · Reviewer_6sPE · 2025-10-15

**Soundness:** 3
**Presentation:** 3
**Contribution:** 1
**Rating:** 2
**Confidence:** 3

**Summary:**

I believe that the authors propose an improvement to TD-MPC2 by adding a regularization term to the policy loss.
The proposed regularizer is a KL-penalty to TD-MPC2's behavior policy instead of the maximum-entropy (or uniform policy) regularizer that we see in standard TD-MPC2.

The authors argue that this is fundamentally improves a bottleneck in value learning/ bootstrapping.
Meaning that the regularization to the behavior policy will make the value bootstraps more accurate to the current policy, and thus reduce the mismatch between the prior or maximum-entropy policy.
This makes sense intuitively, and the authors also present theoretical results to back this.

The experiment section is quite extensive and also includes a didactic result in figure 1, showing that value mismatch is improved by the authors' method.
Overall, the new method typically improves upon the baselines.

**Strengths:**

- The empirical results look very promising for such a simple change to the base TD-MPC2. I think it will be useful to other people that this finding gets advertised around so that anyone using TD-MPC2 at least considers the use of a behavior-policy regularizer over the maximum-entropy regularizer.
- Clear visualization of the methods' effect in figure 1.

**Weaknesses:**

- Although the theoretical contributions support the intuition of the policy-value mismatch and error accumulation, the results are detached from the methodological contribution that the authors make.

- The document often discusses the problem of value overestimation, however, reading more between the lines, and looking at the methodological contribution. The actual problem being discussed seems to be a value mismatch, but this gets wrongly advertised.

- The methods section is written somewhat murky, mixing in many details of TD-MPC2, making it difficult to isolate what the authors did/ contribute. Thus, I might have underreported the authors contribution; if so, correct me, but this should also be more apparent in the text.

- Most results reported only utilize 3 or 5 seeds per experiment and CIs. Some of the results are thus noisy and have overlapping confidence bands, see e.g., ablations in figure 4, or in the appendix figure 11. More seeds should also smoothen out the instable spikes in e.g., figure 2.

- I am confused why DreamerV3 is a baseline in figure 2? It is not a MPC-method. Of course its inclusion does not hurt, but I also don't see why you would bother if better baselines exist which do use some form of MPC. For example, stochastic Alpha-/MuZero (MCTS based planning instead of MPPI).

*Very minor comment:* The paper is decently written, but could use some minor spelling/ grammar revisions for improved flow (can use LLM for this).

*My summary/ verdict:*
Overall I think this is a nice paper with promising results that also align with intuition/ previous theoretical findings.
However, the contribution (in my opinion) feels too much like a work-in-progress.
As I mentioned, I base this on the detached theory with respect to the proposed method.
Furthermore, although the method gets advertised as reducing value overestimation, I believe it really is about reducing a mismatch in learning and using the policy and value neural networks, which is quite different from overestimation.
Still, I think this paper will probably have a good audience at a workshop, so I would advice the authors to consider that instead and revise/ expand where necessary based on this and other reviews.

**Questions:**

1. How is your method related to MPO? Since you substitute the uniform policy for the MPPI policy in the KL-regularizer, is your method still soft-optimal? Or will this regularizer slowly improve as learning continues, ultimately contracting to a stochastically optimal policy?

2. Related to 1), although the use of the regularizer to the MPPI policy improves the mismatch of value and policy learning, would it not have also made sense to use a better value estimator instead (e.g., Retrace where $\mu$ is used as the behavior policy term)? In this way, value learning is more closely aligned with the maximum-entropy objective.

3. Taken 1) and 2) together, if I'm correct that your method is more like MPO rather than SAC (i.e., no longer tracking soft-optimality), how does this change in objective skew the results that you present? Is there a way to create a more fair comparison?

4. Is there a way to extend your theoretical results to modulate the regularization in Eq.10? This would create a strong contrast with the baseline TD-MPC2 while also validating your results.

---

> ### Author Response · Authors · 2025-11-19
>
> We thank the reviewer for the careful reading and for highlighting the strengths of the empirical results and the intuition behind our analysis.  First, we would like to clarify the problem we studied and our main contributions.
>
> > The actual problem being discussed seems to be a value mismatch, but this gets wrongly advertised. Theoretical results are detached from the methodological contribution.
> >
>
> Value overestimation is a result of a mismatched policy in plan-based MBRL. To the best of our knowledge, prior work in MBRL does not systematically study this mismatch nor its compounding effect on value approximation error. The proposed regularizer is **motivated directly by this analysis.**
>
> We agree that the mismatch between the planner policy and nominal policy is the root cause, but the major issue is still **value overestimation**. As we analyzed in section 3, we reveal the following dilemma in off-policy RL is severe in plan-based MBRL despite their strong empirical performance: mismatch → OOD state-action pair → overestimated value targets → error accumulation. Our empirical observations in Figures 1 and 7 show that even in high-dimensional continuous control tasks (h1hand_run and h1hand_slide), the planner's performance is poor, yet the value estimates are exceptionally high, suggesting function approximation bias or extrapolation error.
>
> > The methods section is written somewhat murky, mixing in many details of TD-MPC2, making it difficult to isolate what the authors did contribute.
> >
>
> We appreciate this feedback and will restructure Section 3.3 to isolate the algorithmic update more explicitly. Due to the page limit, we defer some details to Appendix D and devote more space to Section 3, which is more critical to our main contribution. We will include more details in the revised version, which will help readers more easily see what is new and how it follows from the mismatch analysis. In addition, we show that this minimalist implementation is effective enough and compares with other formulations that enforce constraints. As the reviewer **KkN4 noted**, this simplicity supports our theoretical analysis.
>
> > Regarding why DreamerV3 is a baseline: Why you would bother if better baselines exist which do use some form of MPC
> >
>
> We agree that DreamerV3 is **not** an MPC planner‐based method and may not be the most conceptually “fair” baseline. We originally intended to show that our method is competitive with model‐based RL pipelines in general. We mainly consider high-dimensional continuous control. MCTS-based planning methods are different since they are searching in a discrete action space, and discretizing a high-dimensional continuous action space (e.g., 61-dof humanoids) would make the search inefficient. Instead, we consider comparing with BMPC [1] in the revised paper, a variant of TD-MPC that leverages expert iteration similar to MuZero-reanalyze [2].
>
> | task | Ours | BMPC |
> | --- | --- | --- |
> | H1hand-run | 865.60 ± 27.3 | 236.0 ± 53.9 |
> | H1hand-slide | 909.78 ± 33.7 | 440.1 ± 25.4 |
> | H1hand-walk | 928.59  ± 5.0 | 672.6 ± 10.4 |
> | H1hand-stand | 952.45 ± 5.2 | 780.0 ± 65.8 |
> | H1hand-sit_hard | 904.51 ± 3.57 | 688.2 ± 46.3 |
> | H1hand-pole | 887.55 ± 105.21 | 739.9 ± 18.0 |
> | H1hand-hurdle | 212.87 ±  1.63 | 197.1 ± 12.1 |
>
> We also reply to the following questions:
>
> 1. Relation with MPO:
>
> There’s no direct relation between our policy objective and MPO, and the “target distribution” has a distinct formulation. MPO performs mirror-descent steps toward an implicit E-step target distribution (a closed-form solution of **trust-region policy improvement**). In comparison, the KL-term is motivated differently to address the off-policy issue in our case. Despite KL-divergence being used as a regularizer, it is introduced to avoid unsafe out-of-distribution queries when bootstrapping the value function, while MPO and other CPI methods use it to ensure stable and reliable local policy updates. This formulation no longer corresponds to tracking a soft-optimal policy. We believe that a different policy optimization formulation is not the core issue.
>
> 1. The possibility of a better value estimator.
>
> We agree that improving value learning is indeed an appealing way to address overestimation. Our base algorithm (TD‑MPC2) already uses Layer Normalization and a cross-entropy loss to help stabilize value updates, yet overestimation still remains. We intend to explore them in future work.
>
> 1. Is there a way to extend your theoretical results to modulate the regularization in Eq.10?
>
> One possible way is to set the divergence margin \epsilon and update \beta autonomously as in MPO. We empirically found that our method is not sensitive to \beta as in Figure 4.b.
>
> References:
>
> [1] Wang, Yuhang, et al. "Bootstrapped model predictive control." *arXiv preprint arXiv:2503.18871* (2025).
>
> [2] Schrittwieser, Julian, et al. "Mastering atari, go, chess and shogi by planning with a learned model."

---

> > ### Comment · Reviewer_6sPE · 2025-11-20
> >
> > Thank your for your rebuttal, see my response.
> >
> > ---
> > I agree that you show that there is overestimation happening in the value functions, this is clearly shown in figure 1. My point remains: while your regularizer "fixes" the overestimation in figure 1, at the moment I cannot say with certainty whether overestimation was the problem to begin with.
> >
> > When you generate learning targets for the action-value function $Q(s, a)$ and policy $\pi(a | s)$ I would imagine that the on-policy value of the agent is better than the prior policy $\pi$. Because the on-policy values actually estimate the behavior of your agent that uses planning, and $\pi$ alone does not use the MPPI planner. So since the value estimates are tracking a better policy, it's likely that they turn out to be larger.
> >
> > And it indeed makes sense to then add a regularizer for learning $\pi$ that is an approximation to the actual on-policy behavior (referring to $\mu$ in Eq. 10). But, this means that $\pi$ is updated to the better distribution in true value, $\mu$, which thus aligns $\pi$ closer to your so called overestimated values.
> >
> > Do you see my point?
> >
> > There is definitely a mismatch in value and policy learning, in your specific setting this turns out to also be an overestimated value. But to argue that the regularizer fixes **overestimation**, that claim I cannot get behind.
> >
> > ---
> > On the value estimator part, I was referring to alternative methods to estimate the value functions $V$ or $Q$. I think you misunderstood what I meant and confused it with value learning.
> >
> > I agree that layer norm or distributional approaches can help achieve greater predictive power, but what I want to know is if your overestimation problem can instead be improved by generating better targets.
> >
> > For example, look at the approach by double Q-learning and how they use two Q functions. Or more recently the clipped double-Q trick from TD3, which takes the minimum of the two Qs at $\max_a \min_i Q_i (s, a)$.
> > Alternatively, you can use Retrace to correct through importance sampling that there is a mismatch between $\mu$ and $\pi$.
> >
> > Can you rerun the baseline TD-MPC2 (without your KL-regularizer!) with the double Q-learning trick? Or Retrace? Or both? Does this fix the problem? If yes, then I would be more inclined to trust your claims. If no, then something else might be happening that we're now missing.

---

> > > ### Author Response · Authors · 2025-11-28
> > >
> > > Thank you for your patience, and we sincerely appreciate the timely follow-up. We believe there is still a misunderstanding regarding what policy the value function is evaluating in Figure 1, and we would like to clarify this precisely.
> > >
> > > > The on-policy value of the agent is better than the prior policy, …, So since the value estimates are tracking a better policy, it's likely that they turn out to be larger.
> > > >
> > >
> > > However, throughout all experiments, the value function is trained to evaluate the nominal policy \pi, as defined by the TD learning objective in Eq. 9. **In Figure 1, the true value is estimated via Monte Carlo sampling with respect to the nominal policy \pi;** that is, we roll out \pi to generate samples rather than the planner policy \pi_H.  We had clearly denoted it as $V^\pi$ and learned value estimate as $\hat{V}$. Importantly, value learning in the TD-MPC family is not evaluating “on-policy value” since the planner policy is non-parameterized and sampling from it is expensive during training (Requires batch_size * n_iteration * n_trajectories * horizon times evaluation of the world model). Thus, the value is acquired through a separate policy iteration [1], the hope is to approximate the optima $V*$ iteratively. **The value function evaluates the nominal policy using off-policy data**, where value function approximation occurs.
> > >
> > > > But, this means that \pi is updated to the better distribution \mu in true value, which thus aligns closer to your so called overestimated values.
> > > >
> > >
> > > Overestimated values are not due to the planner policy being better than the nominal policy, but erroneously assigning a high value to states that resemble. For instance, in the H1-hand-run environment, compared with Figures 1 and 2, there is clear evidence that even the planner policy is poor for the TD-MPC2 baseline. However, the learned value function grows extremely high. Another piece of evidence is that simply adding \pi’s rollout data into the replay buffer can mitigate this overestimation (row 2 of Figure 1). And by comparing the numerical results between the first and second rows, we see that an increase in ground truth value and a decrease in learned value occur simultaneously. This clearly indicates that we are facing an off-policy issue, as discussed in [2, 3, 4].
> > >
> > > > I agree that layer norm or distributional approaches can help achieve greater predictive power, but what I want to know is if your overestimation problem can instead be improved by generating better targets.
> > > >
> > >
> > > We appreciate this observation. Indeed, generating more stable targets helps mitigate the issue. We already adopt a clipped double-Q variant (via ensemble sampling of two Q-functions and taking their minimum) in the TD-MPC2 baseline, but substantial overestimation still persists. This is expected: CDQ reduces some positive bias but does not eliminate **extrapolation error**, especially when the value function is queried at out-of-distribution state-actions.
> > >
> > > We agree that Retrace's truncated importance-weighted multi-step bootstrapping offers a promising direction[5], since it safely corrects off-policy trajectories without querying new actions that could diverge far. Given the long training time of TD-MPC family, we will do our best to include preliminary results.
> > >
> > > References:
> > >
> > > [1] Munos, Rémi. "Error bounds for approximate policy iteration." *Proceedings of the Twentieth International Conference on International Conference on Machine Learning*. 2003.
> > >
> > > [2] Fujimoto, Scott, Herke Hoof, and David Meger. "Addressing function approximation error in actor-critic methods." *International conference on machine learning*. PMLR, 2018.
> > >
> > > [3] Fujimoto, Scott, David Meger, and Doina Precup. "Off-policy deep reinforcement learning without exploration." *International conference on machine learning*. PMLR, 2019.
> > >
> > > [4] Kumar, Aviral, et al. "Stabilizing off-policy q-learning via bootstrapping error reduction." *Advances in neural information processing systems* 32 (2019).
> > >
> > > [5] Munos, Rémi, et al. "Safe and efficient off-policy reinforcement learning." *Advances in neural information processing systems*29 (2016).

---

### Official Review · Reviewer_87XR · 2025-10-27

**Soundness:** 3
**Presentation:** 3
**Contribution:** 2
**Rating:** 4
**Confidence:** 4

**Summary:**

This paper proposes a modification to the TD-MPC2 algorithm, aiming to reduce the off-policyness of its policy iteration updates. The authors identify that in TD-MPC2, the actor policy can diverge from the planner, i.e. the data-collection policy, leading to off-policy value estimation errors and overestimation bias. To address this, the paper introduces a simple regularization term that penalizes divergence between the actor policy and the the data-generating policy, effectively aligning the learned policy with past planner distributions. The resulting method, TD-M(PC)$^2$, is conceptually simple but empirically effective. It improves performance on several continuous-control benchmarks, including DMC and HumanoidBench, while adding minimal computational or implementation overhead.

**Strengths:**

- The proposed modification is algorithmically minimal but addresses a real and well-known issue in all of RL: overestimation errors in highly off-policy settings.
- Across the tested domains, TD-M(PC)$^2$ consistently improves upon TD-MPC2 in both sample efficiency and stability. The magnitude of improvements is sometimes very large on the complex humanoid-bench environments.
- The paper gives an intuitive argument connecting reduced off-policyness with less value overestimation. The introduction of a KL regularization term between the actor and stored planner policies is natural and easy to follow.

**Weaknesses:**

**Novelty**:

The biggest weakness of this contribution, in my view, is its novelty. The proposed regularization term closely resembles known conservative regularization terms from the offline RL literature. The behavior cloning (BC) penalty term suggested by Fujimoto et al. - a widely used trick in offline and off-policy RL - performs nearly on par with the shown algorithm. The conceptual novelty thus lies mainly in the application context (TD-MPC2) rather than in the underlying idea.

**Experiments**

It is dissatisfying that the authors use a different set of baseline algorithms in their experiments. While the shown curves do tend to correspond roughly to known results for these algorithms, the fairness of the experimental comparisons is somewhat unclear. For example, why is Dreamer-v3 not used as a baseline for the DMC suite, the main suite it has been optimized for (and is known to perform well on)? In contrast, the results of Dreamer-v3 seem to match roughly what is reported by Sferrazza et al. in the humanoid bench, but notably this is not the original implementation of Dreamer-v3 and it is unclear how well this model was tuned in the context of this benchmark paper. More generally, it would be empirically more valuable if all algorithms were tested on all benchmarks and underwent a comparable tuning pipeline.

**Others**:

- The algorithm regularizes the current actor toward the planner that collected the data. This ensures conservative updates but effectively ties the learned policy to a historic dataset. This introduces the pathology that the algorithm can not solve problems optimally if the data-collection policy was suboptimal, even in the limit of infinite data. A natural alternative - which is not explored in this work - would be to regularize the current actor toward the current planner rather than historical ones. Algorithmically, this is more difficult because one would have to plan or find clever alternatives to acquire up-to-date properties of the planning policy, but I would find this alternative algorithm insightful.

- While the paper is easy to follow conceptually, it has several language and stylistic issues that detract from readability. Minor grammatical errors and inconsistent notation appear throughout.

Scott Fujimoto and Shixiang Shane Gu. A minimalist approach to offline reinforcement learning.
Advances in neural information processing systems, 34:20132–20145, 2021.

Carmelo Sferrazza, Dun-Ming Huang, Xingyu Lin, Youngwoon Lee, and Pieter Abbeel. Humanoid-
bench: Simulated humanoid benchmark for whole-body locomotion and manipulation. arXiv
preprint arXiv:2403.10506, 2024.

**Questions:**

- Could you discuss the potential of regularizing toward the current planner $\pi_{H,k}$ rather than the historical, data-collection planner? Would this be computationally feasible or beneficial?
- Did you implement a hyperparameter tuning protocol and could you run a consistent set of baselines, tuned accordingly?

---

> ### Author Response · Authors · 2025-11-19
>
> Thank you for your constructive feedback. First, we would like to address the concern regarding novelty. A principal contribution of our work is that we **identify and quantify a previously overlooked value-overestimation issue** in the state-of-the-art plan-based model-based RL framework. We then provide a **theoretical analysis** showing how this issue becomes a bottleneck in high-dimensional continuous control problems. Our paper is not rooted in the adaptation of policy constraints per se. Rather, the constraints we use emerge *naturally* from the structural issue we analyse. As shown in Appendix E, our ablation experiments and comparisons to other conservativeness-enforcement methods demonstrate that a simple reverse-KL penalty is sufficient to mitigate the overestimation and improve performance. Thus, we view the simplicity of the mechanism not as a lack of novelty, but as a **pragmatic validation** of our theoretical insight.
>
> Below, we address the following concerns and questions.
>
> 1. Suboptimal in policy regularization.
>
> We agree that directly mimicking up-to-date behavior policy with higher quality will certainly be beneficial, as shown in expert iteration algorithms.  We consider two major reasons: 1) Computation efficiency, the non-parameterized planner policy is hard to estimate, while re-planning during policy learning is not feasible (we will explain under the next question); 2) Effectiveness: The regularization term is included to reduce out-of-distribution queries, thus **mitigating value error**. Importantly, **what truly matters is the planner $\pi_H$’s performance**; the nominal policy is introduced mainly to acquire a value function for the planning procedure. Despite the constraints on the nominal policy $\pi$, **we do not enforce policy constraints on the planner policy $\pi_H$**. Thus, unlike online trust-region methods and offline policy-constraint techniques for model-free RL, the $\pi_H$’s performance is not strictly hinged on the behavior policy $\mu$. In addition, $\pi_H$ is both theoretically and empirically superior to the nominal policy throughout the training because of the inference time planning, and the performance gap is not marginal. Thus, most of the experiences will not significantly hinder the policy improvement, and the benefit of stable policy evaluation (value learning) prevails in practice (similar to the intuition in MuZero)
>
> 2. Potential of regularizing toward the current planner rather than the historical collected data.
>
> Ideally, regularizing the policy toward the current planner or directly bootstrapping the planner will mitigate the policy mismatch and consequently value overestimation as well. As you mentioned, due to the non-parameterized nature of the planner policy, it's hard to acquire an exact estimation. Meanwhile, replanning on batched states is computationally infeasible (planning at inference time already requires 512 parallel rollouts per iteration to achieve good performance). One alternative is only to replan a small portion in the buffer (reanalyze ratio of 0.8% in [2]). Despite its empirical effectiveness, it in fact leverages a large portion of stale experience, which is further evidence supporting our analysis identifying the core bottleneck. We will also include empirical results in revised paper.
>
> 3. Hyperparameter tuning protocol.
>
> We did not use a separate hyperparameter tuning protocol. To ensure fairness and reproducibility across benchmarks, we followed the practice used in prior MBRL works [1]: we applied **a single **set of hyperparameters** for our method across all tasks and used baseline hyperparameters exactly as reported in their original papers or official implementations. The idea is to avoid overfitting any individual method to specific environments or tasks. However, we observe that hyperparameter tuning improves empirical performance in some tasks (e.g., Table 2), but our method is not sensitive to hyperparameter choice across most tasks. We will clarify this choice in the revision.
>
> [1] Hansen, Nicklas, Hao Su, and Xiaolong Wang. "Td-mpc2: Scalable, robust world models for continuous control." *arXiv preprint arXiv:2310.16828* (2023).
>
> [2] Wang, Yuhang, et al. "Bootstrapped model predictive control." *arXiv preprint arXiv:2503.18871* (2025).

---

> > ### Comment · Reviewer_87XR · 2025-11-26
> > **Response to Authors**
> >
> > Thank you for the responser to my review, I appreciate the response and clarifications by the authors.
> >
> > - Novelty: I believe the diagnosis of value-overestimation in the specific context of TD-MPC is useful, but the novelty is limited to this very specific application context. At this point it is mainly a matter of wording, but the general failure mode of value overestimation and the solution of regularizing policies towards imitation are known in general; This is why "previously overlooked" still reads a bit too strong for me. That being said, I don't consider this a deal-breaker for publication, but in my opinion it raises the standards for the empirical analysis.
> > - Policy improvement: I see that the planner planner policy itself remains unconstrained and may be more performant than a nominal policy in most cases. However, I don't think this will, by itself, mitigate the issue of limiting policy improvement through the proposed regularization. This should very much depend on the data collection setting, e.g., if this algorithm were to be used very off-policy or even offline, I believe it wouldn't be able to perform "trajectory-stitching" but instead would be tied to the data-generating policy. I appreciate that the authors are planning an additional ablation analysis to this end.
> > - Fairness/tuning: I tend to disagree that reusing single hyperparameter settings, especially for novel benchmarks, ensures fairness. Instead I really suggest devising a uniform tuning process with comparable budget across all methods and complete baseline coverage (notably Dreamer-v3 on DMC).
> >
> > As I mentioned I above, I believe the novelty of the contributions to be somewhat limited to this very specific application context, which themselves also do not require significant deviation from known solutions. For this reason, I would put more emphasis on the empirical investigation and its methodological thoroughness. I appreciate that the authors are planning to include additional ablation studies, but to increase my score, I would really prefer a matched tuning pipeline for all methods (even a modest sweep) and including all baselines on all benchmarks. I am therefore inclined to keep my current score.

---

### Official Review · Reviewer_KkN4 · 2025-11-02

**Soundness:** 4
**Presentation:** 4
**Contribution:** 3
**Rating:** 8
**Confidence:** 3

**Summary:**

This paper identifies that persistent value overestimation caused by structural policy mismatch between the MPC planner's H-step lookahead policy and the nominal policy is a bottleneck in plan-based model-based reinforcement learning (MBRL), especially in high-dimensional tasks, and proposes the TD-M(PC)² algorithm which incorporates a distribution-constrained conservative policy update to mitigate out-of-distribution queries, reduce value overestimation, and significantly improve performance over baselines while being seamlessly integrable into existing plan-based MBRL pipelines with negligible additional computational overhead.

**Strengths:**

- The paper makes a distinct original contribution by identifying structural policy mismatch (between MPC planners’ exploration policy and nominal policies for value learning) as the root cause of persistent value overestimation in plan-based MBRL.

- The work demonstrates high methodological and empirical quality. Theoretically, it provides rigorous theorems (Theorems 3.1–3.3) to quantify value approximation error, performance gaps, and distribution shifts, with detailed proofs in appendices. Empirically, it validates on diverse benchmarks, compares against strong baselines to confirm the proposed components’ effectiveness.

- The paper is well-structured and clear.

- The paper demonstrates strong logical coherence, effectively integrating theoretical analysis with experimental observations to present its core arguments and the research problems it aims to address.

**Weaknesses:**

- In the section of *Policy Iteration with Reduced OOD Query*, the approach adopted in this paper aligns more with existing design ideas of policy constraints from other directions, without obvious distinctive innovations. However, this part is not the core focus of the study and therefore has no adverse impact on the overall quality and validity of the research.

**Questions:**

- Due to space limitations, the introduction to the *ABLATION STUDY* is not detailed enough. It would be more comprehensive if the methods in Figure 4 could be introduced in the appendix.

- Could you please elaborate on the following points regarding the section *Appendix E: Conservative Policy Learning*? First, what types of policy constraint methods have been attempted? Second, has there been an analysis of the reasons why certain methods are not applicable under the current setting?

---

> ### Author Response · Authors · 2025-11-19
>
> We sincerely appreciate your positive assessment of the paper’s contribution and constructive comments. Below, we would like to address your concerns and questions.
>
> 1. Relationship to Existing Policy-Constraint Ideas
>
> We agree that the distribution-constrained update shares high-level intuition with existing policy-constraint methods (e.g., in offline RL). Our goal was not to propose a new constraint family per se, but to **ground the need for such a constraint in a new structural diagnosis,** which is the persistent overestimation caused by planner–policy mismatch in MBRL. We used a constraint that is analytically justified to reduce mismatch-induced  and both empirically effective and easy to implement. We also include empirical results for several alternative constraint formulations in the ablation section and Appendix E. These variants, which share similar intuition, generally help mitigate this issue and improve baseline performance in high-dimensional tasks. However, the RKL objective we use is the most stable and consistently strong in practice.
>
> 2. Details on methods included in Figure 4.
>
> We consider two alternative formulations of policy constraints. **TDMPC-BC** replaces the log-likelihood objective with the vanilla BC objective as follows:
>
> $$
> \mathcal{J} = \mathbb{E}_{a\sim\pi(\cdot|s)}\left[Q(s, a) + \beta \mathbb{E}_{a'\sim\mu(\cdot|s)}(a-a')^2 - \alpha\log\pi(a|s)\right]
> $$
>
> Another variant, **TDMPC-MC-Mean**, only considers matching the mean action of the behavioral policy.
>
> $$
> \mathcal{J} = \mathbb{E}_{a\sim\pi(\cdot|s)}\left[Q(s, a) + \beta \mathbb{E}_{\mu'\sim\mathcal{B}, a'\sim\mu'(\cdot|s)}(a-\mathbb{E}[a'])^2 - \alpha\log\pi(a|s)\right]
> $$
>
> 3. More elaboration on points regarding the section *Appendix E.*
>
> Offline RL methods are effective in addressing conservatism, but due to the usage of a short-horizon planner, overly conservative or underestimated values may hinder online exploration. Based on this intuition, we avoid directly regularizing the value function (e.g., CQL). We also implemented AWAC-style policy extraction strategy using forward-KL objectives; Despite this variant also mitigating value estimation, the empirical performance is not as good as TD-M(PC)^2. We assume this is due to the FKL formulation mainly encouraging “zero-avoidance” behavior, but in cases, mode seeking could be important in online RL. The detailed discussion can be found in Appendix D.

---

### Official Review · Reviewer_7WMK · 2025-11-02

**Soundness:** 3
**Presentation:** 4
**Contribution:** 3
**Rating:** 4
**Confidence:** 3

**Summary:**

This work is in the area of planner-Based Model-based Reinforcement Learning. They look at a recently proposed method TD-MPC2 and propose that the performance of this method can suffer due to the overestimation bias of the value function. They show empirically and theoretically that this overestimation comes from the divergence of the planning policy that collects the data and the sampling policy that the value function evaluates.

The authors propose to alleviate this by adding a KL divergence term in the loss of the learned policy in order to keep the learned policy close to the behaviour policy induced by the replay buffer. Experiments show improvements in performance across most of the benchmark tasks.

**Strengths:**

I really liked the way the authors explained the problem, showed evidence for it, and proposed a simple approach to solve it. The writing was clear and flowed well from section to section. Figure 1 was a great illustration for their point and the effectiveness of their approach overall. The theoretical results were explained well, and supported the proposal of the paper. Experimental results also supported their points and showed improvement over the main baseline given the simple and practical solution proposed. Overall, this is a strong paper.

**Weaknesses:**

The paper’s proposed idea is very similar to (Wang et al. 2025), but this is not explained very well in the paper. The authors mention this in L934-936 but I believe their explanation is not complete: BMPC uses the planner’s data as the expert to perform imitation learning, meaning they use almost the same objective as the current paper, with the difference that they do not add the KL objective to the policy loss but perform it as a separate step (i.e. imitation learning), and they also filter the replay buffer data to keep it more up-to-date. I  believe these differences should be expanded on in the paper. I think BMPC should be added as a baseline as well.

I think the exposition of the approach itself needs a bit more space as well. Specifically the section starting at L293. The practical implementation of the objective in Eqn (8) is briefly talked about but the policy loss in Eqn (10) does not show how $\log \mu$ is lower-bounded. If some of the exposition in Appendix D were added here (e.g. L1043 - 1053) it would be a lot clearer. I understand that there are space limitations but this is a core part of the proposed approach.

On a more minor side, some of the notation used can be confusing.
- I think using $\pi_k$ and $\pi_{H,k}$ for the two different policies is too similar, perhaps $\Pi_{H,k}$ for the planner policy might make it more readable, or using $\pi_{\theta, k}$ for the learned policy to indicate there are parameters that are learned
- In Theorem 3.1, $\epsilon_k$ is the approximation error of the value function, but we also see $\epsilon_{m,k}$ and $\epsilon_{p,k}$. Perhaps it would be clearer if the approximation error of the value function was $\epsilon_{v,k}$ to make it clearer that k refers to the iteration.
- L310: multi-variant Gaussian -> mixture of Gaussians?
- L317: “s” for the threshold is immediately used to mean state in Eq (10) and elsewhere.
- Sometimes the replay buffer is denoted as $\mathcal{D}$ and sometimes as $\mathcal{B}$ -- e.g. Eqn (8) vs. Eqn (10).
- L403 -- “up-to-date (UTD)” → “update-to-data (UTD)”

**Questions:**

- (L308) why do you consider reverse KL and not forward KL?
- In L311: what does $\bar{\pi}$ refer to?

---

> ### Author Response · Authors · 2025-11-19
>
> We thank the reviewer for the prompt feedback. We carefully went over the suggestions regarding the notations and will apply these changes in the revised version to improve readability and notation consistency. Below, we provide our responses to each of the questions, and we will address these in the revised version.
>
> 1. Compare against BMPC.
>
> While BMPC also includes an imitation objective that aligns the nominal policy towards the planner during policy improvement, we argue that this similarity is merely methodological and that BMPC is distinctively motivated.  BMPC considers approximating an ideal case of directly bootstrapping the planner policy. This is based on the observation that the nominal policy tends to underperform the planner policy, and bootstrapping a better policy of the planner will improve value learning. Since the planner policy is non-parametrizable, they adopted “lazy analysis” to update the buffer and imitate this updated behavior. However, this is not simply the problem of value mismatch, but overestimation originating from bootstrapping off-policy. As demonstrated in Figure 1, even in tasks where the planner policy is poor (*h1hand-run and h1hand-slide*), the learned value estimation is erroneously high. We argue that the bottleneck is the approximation error that results in value overestimation. BMPC’s good performance is predictable under this explanation: By bootstrapping a policy that is close to the behavior policy (planner), it mitigates policy mismatch, thus reducing out-of-distribution queries, leading to a more accurate value estimation. This is essential for policy iteration algorithms to perform stable improvement. Due to limited time and resources, we first provide initial comparison results on HumanoidBench. and will include BMPC as a baseline in the revised version. (mean and std across 3 seeds)
>
> | task | Ours | BMPC |
> | --- | --- | --- |
> | H1hand-run | 865.60 ± 27.3 | 236.0 ± 53.9 |
> | H1hand-slide | 909.78 ± 33.7 | 440.1 ± 25.4 |
> | H1hand-walk | 928.59  ± 5.0 | 672.6 ± 10.4 |
> | H1hand-stand | 952.45 ± 5.2 | 780.0 ± 65.8 |
> | H1hand-sit_hard | 904.51 ± 3.57 | 688.2 ± 46.3 |
> | H1hand-pole | 887.55 ± 105.21 | 739.9 ± 18.0 |
> | H1hand-hurdle | 212.87 ±  1.63 | 197.1 ± 12.1 |
>
> 2. More exposition of the method.
>
> We agree that the method section should include more details. Regarding the selection of the lower bound of \log\mu (where \mu is the behavior policy stored in buffer), since density estimation is computationally intensive, we consider sampling the lower bound acquired through Jensen’s inequality:
>
> $$
> \log\mu(a|s) = \log \mathbb{E}[\mu'(a|s)] \geq \mathbb{E} [\log\mu'(a|s)]
> $$
>
> Where \mu’ is the planner policy of past rollouts (denoted as \hat{\pi}_H) and a Gaussian distribution, we stored its mean and standard deviation alongside transition (s, a, r, s’, d) during exploration. We will add related discussions in Appendix D in the main paper.
>
> 3. Why consider reverse KL but not forward KL
>
> We provide analysis and ablations on this question in Appendix D.1 and Appendix E (Figure 11). We implemented the FKL-style policy objective based on AWAC [1]. Empirically, we found that the FKL variant (AWAC-MPC) performs worse than the RKL variant (Ours) with slower convergence and lower asymptotic performance. This observation aligns with [2]; we assume this is due to FKL's zero-avoidance behavior leading to less effective policy improvement, given that we are using a unimodal Gaussian (nominal policy) to match a mixture of Q-weighted history-planner policies.
>
> 4. In L311: what does \hat{\pi}_H refer to?
>
> \hat{\pi}_H(\cdot|s) refers to the planner policy from past iterations, which is sampled from the replay buffer. We provide details in Appendix D.1.
>
> Reference:
>
> [1] Nair, Ashvin, et al. "Awac: Accelerating online reinforcement learning with offline datasets." *arXiv preprint arXiv:2006.09359*(2020).
>
> [2] Park, Seohong, et al. "Is value learning really the main bottleneck in offline RL?." *Advances in Neural Information Processing Systems* 37 (2024): 79029-79056.

---

### Public Comment · ~Zhongben_Gong1 · 2025-11-21
**Adding a comparison with BOOM may be helpful**

Thank you to the authors for the solid work and for the detailed rebuttal. I appreciate the clarifications provided. I would like to gently point out that the proposed method appears to share certain similarities with Bootstrap Off-policy with World Model (BOOM) [1]. In particular, both approaches incorporate an alignment-style regularization term to encourage consistency between the learned policy and the MPPI.

It would be helpful if the authors could further clarify the conceptual or methodological differences between their alignment regularization and the one used in BOOM. Such a clarification would strengthen the paper's contribution and positioning. Additionally, if feasible, including BOOM as a baseline in the empirical comparison would provide a more complete understanding of the advantages and limitations of the proposed method.

[1] Zhan, Guojian, et al., “Bootstrap Off-policy with World Model,” NeurIPS 2025

---

> ### Public Comment · ~Guojian_Zhan1 · 2025-11-21
>
> Hi, Zhongben, thank you for bringing up BOOM [1]. As the author of that work, I have also taken note of this paper, and I acknowledge that BOOM and TDM(PC)$^2$ are concurrent papers in this year motivated by the same planner-policy distribution mismatch issue, which resembles offline RL. The key distinction lies in the alignment mechanism: BOOM follows an AWAC approach, utilizing Critic-guided weights and likelihood-free forward KL for alignment. TDM(PC)$^2$ seems to provide a more detailed experimental and theoretical analysis of value overestimation. Empirically, it follows the simple yet effective TD3+BC paradigm, employing a reverse KL loss to achieve strong performance and demonstrably accurate value estimation. I would be very grateful if BOOM could be cited.

---

### Note · Authors · 2025-12-03

I have read and agree with the venue's withdrawal policy on behalf of myself and my co-authors.